# Phosphorylation of *Xenopus* M18BP1 governs centromeric localization and CENP-A nucleosome assembly

Rae R Brown [ID], Jacob P Schwartz [ID], Lyin Ghadri [ID] & Aaron F Straight [ID] [✉]

## Abstract

**Eukaryotic chromosome segregation requires attachment of chromosomes to microtubules through the kinetochore so that chromosomes can align and move in mitosis. Kinetochores assemble on the centromere, which is epigenetically defined by the histone H3 variant CENtromere Protein A (CENP-A). During DNA replication, CENP-A is equally divided between replicated chromatids, and new CENP-A nucleosomes are re-assembled during the subsequent G1 phase. How cells regulate the cell cycle timing of CENP-A assembly is a central question in the epigenetic maintenance of centromeres. CENP-A nucleosome assembly requires the Mis18 complex (Mis18α, Mis18β, and M18BP1), whose localization to centromeres occurs between metaphase and G1. Here, we define a new regulatory mechanism that works through phosphorylation of *Xenopus laevis* M18BP1 between metaphase and interphase. Phosphorylation disrupts binding of M18BP1 to CENP-A nucleosomes in metaphase, and when relieved, enables M18BP1 binding to CENP-A nucleosomes in interphase. We show that this phosphorylation-dependent mechanism regulates CENP-A nucleosome assembly. We propose that the phospho-regulated binding of M18BP1 to CENP-A nucleosomes restricts new CENP-A assembly to interphase.**

**Keywords** Centromere; Phosphorylation; Nucleosome Assembly; Cell Cycle Regulation
**Subject Categories** Cell Cycle; Chromatin, Transcription & Genomics

## Introduction

All eukaryotic cells undergo the fundamental process of cell division, wherein each daughter cell receives an equal chromosome complement. Central to this process is the centromere, the region of the chromosome to which the mitotic spindle attaches for accurate chromosome segregation. The centromere is epigenetically defined by the histone H3 variant CENtromere Protein A (CENP-A) (Warburton et al, 1997; Palmer et al, 1987; Sullivan et al, 1994;

Earnshaw and Rothfield, 1985; Palmer et al, 1991; Meluh et al, 1998; Takahashi et al, 2000; Henikoff et al, 2000). Loss of centromere identity and disruption of CENP-A result in aneuploidy, chromosome instability, and cell death (Stoler et al, 1995; Blower and Karpen, 2001; Howman et al, 2000; Régnier et al, 2005; Goshima et al, 2003).

With each round of cell division, CENP-A nucleosomes are split between daughter cells, and new CENP-A nucleosomes must be assembled in order to faithfully maintain the centromere. H3 nucleosomes are assembled during the S phase of the cell cycle and coupled with DNA replication (Groth et al, 2007; Ramachandran and Henikoff, 2015; Worcel et al, 1978). However, in many metazoans, CENP-A nucleosome assembly is decoupled from DNA replication and occurs during late telophase/early G1 (Jansen et al, 2007; Maddox et al, 2007; Moree et al, 2011; Bernad et al, 2011; Silva et al, 2012). While this timing is controlled through the cell cycle by oscillations in phosphorylation and proteolysis, the full repertoire of substrates and their coordination in controlling CENP-A assembly is not fully understood.

Holliday Junction Recognition Protein (HJURP) is a histone chaperone for the CENP-A:H4 dimer that is necessary for CENP-A nucleosome assembly. However, HJURP cannot localize to the centromere and bind to CENP-A nucleosomes on its own (Dunleavy et al, 2009; Foltz et al, 2009). In frogs (*Xenopus laevis*) and chickens (*Gallus gallus*), both the Mis18 complex (a hetero-octameric complex composed of Mis18α tetramer, a Mis18β dimer, and two molecules of M18BP1) and CENtromere Protein C (CENP-C) bind directly to CENP-A nucleosomes. M18BP1 and CENP-C bind to HJURP, thereby serving as adapters to localize HJURP:CENP-A:H4 to centromeres and facilitate the deposition of the CENP-A:H4 histone dimer. However, while CENP-C is constitutively localized to the centromere, the Mis18 complex only localizes in interphase/G1 (Moree et al, 2011; French et al, 2017; Carroll et al, 2010; Watanabe et al, 2019; Ariyoshi et al, 2021; Jiang et al, 2023). In humans, the Mis18 complex cannot bind to CENP-A nucleosomes. Instead, the Mis18 complex binds to CENP-C to facilitate CENP-A assembly (McKinley and Cheeseman, 2014; Nardi et al, 2016; Fujita et al, 2007; Maddox et al, 2007; Pan et al, 2019; Moree et al, 2011; Dambacher et al, 2012).

Understanding how the Mis18 complex directly engages CENP-A nucleosomes and how that interaction changes between metaphase and interphase is important for understanding the cell

Department of Biochemistry, Stanford University School of Medicine, Stanford, CA, USA. ✉E-mail: astraigh@stanford.edu

cycle regulation of CENP-A assembly. *X. laevis* and *G. gallus* M18BP1 contain a CENP-C motif, which is absent in humans. This motif enables binding to CENP-A nucleosomes via the same mechanism used by CENP-C (Hori et al, 2017; French et al, 2017; Kral, 2015; Jiang et al, 2023). *X. laevis* is allotetraploid with two subgenomes, and as such it has two isoforms of M18BP1: M18BP1-S and M18BP1-L. Both isoforms have the conserved CENP-C motif and bind to CENP-A nucleosomes in vitro (French et al, 2017). In interphase egg extract, both M18BP1-L and M18BP1-S bind to CENP-A nucleosomes as part of the Mis18 complex to facilitate CENP-A assembly. However, in metaphase egg extract, the Mis18 complex dissociates, and neither M18BP1 isoform binds to CENP-A nucleosomes. M18BP1-S localizes to the centromere in metaphase by binding to CENP-C, while M18BP1-L does not localize to the centromere. It remains unclear what disrupts the M18BP1-CENP-A nucleosome interaction such that both M18BP1 isoforms are no longer bound to CENP-A nucleosomes in metaphase.

CENP-C, the Mis18 complex, and HJURP are regulated by mitotic kinases, and in particular Polo-like Kinase 1 (Plk1) and Cyclin-dependent kinase 1 (Cdk1) (McKinley and Cheeseman, 2014; Silva et al, 2012; Pan et al, 2017; French and Straight, 2019; Stankovic et al, 2017; Spiller et al, 2017; Müller et al, 2014). Cdk1 phosphorylation triggers the disassembly of the Mis18 complex and disrupts HJURP localization to centromeres by blocking its Mis18 complex and CENP-C binding (McKinley and Cheeseman, 2014; Pan et al, 2017; Spiller et al, 2017; Müller et al, 2014; French et al, 2017; Pan et al, 2019; French and Straight, 2019; Stankovic et al, 2017; Wang et al, 2014; Stellfox et al, 2016).

In this work, we examine the cell cycle regulation of CENP-A assembly in *X. laevis*. We find that the mitotic phosphorylation of a conserved residue, serine 760 (S760), in the CENP-C motif of M18BP1-L prevents binding to CENP-A nucleosomes, thereby restricting CENP-A assembly to interphase. Consistent with this, phosphomimetic mutation of the S760 residue of M18BP1-L to aspartic acid (S760D) in interphase extracts, when CENP-A is normally assembled, disrupts M18BP1-L localization to centromeres and inhibits CENP-A assembly. In metaphase egg extract, a non-phosphorylatable mutant of M18BP1-L, S760 mutated to alanine (S760A), is unable to localize to the centromere, indicating that additional mechanism(s) prevent metaphase binding of M18BP1-L to CENP-A nucleosomes.

# Results

## M18BP1-L binding to CENP-A nucleosomes is disrupted by a conserved mitotic phosphorylation

To identify residues in M18BP1-L that regulate the timing of its centromere localization, we previously mapped mitotic phosphorylation sites in its CENP-A binding domain (Fig. EV1B). Of these sites, mutation of the Cdk1 consensus phosphorylation sites did not disrupt the binding of M18BP1-L to CENP-A nucleosomes in vitro (French et al, 2017). Thus, we focused our analysis on the next most highly phosphorylated residue, serine 760 (S760). We verified the cell cycle specificity of this phosphorylation by mapping phosphorylation sites for both M18BP1-L[747-944] and M18BP1-S[750-917] in interphase and metaphase with mass spectrometry (Fig. EV1A).

We found no evidence of M18BP1-L S760 phosphorylation in interphase egg extract (Fig. EV1B). We found that M18BP1-L S760 is phosphorylated in metaphase; however, we could not determine the phosphorylation of the homologous residue in M18BP1-S (serine S772) because we could not detect the peptide in which it was located in the mass spectrum (Fig. EV1C).

Interestingly, the phosphorylated S760 in M18BP1-L is located near the conserved arginine anchor identified as being important for the affinity of M18BP1[ggKNL2] for CENP-A nucleosomes via the H2A/H2B acidic patch (Jiang et al, 2023). This recent cryo-EM structure of *G. gallus* M18BP1[KNL2] bound to the CENP-A nucleosome showed that M18BP1[KNL2] interacts with the C-terminal tail of CENP-A, the acidic patch formed by H2A and H2B, and the L1 loop of CENP-A (Jiang et al, 2023). The sequence of the CENP-A nucleosome binding domain of *X. laevis* M18BP1-L is conserved in *G. gallus* M18BP1[KNL2] and other species (Kral, 2015; Jiang et al, 2023; French et al, 2017) (Fig. 1A). We generated an AlphaFold-based prediction of the *X. laevis* M18BP1-L bound to the *X. laevis* CENP-A nucleosome, which showed that the same contacts are predicted in the *X. laevis* model that were shown to occur in the *G. gallus* structure (Figs. 1B and EV2). We hypothesized that phosphorylation of the S760 site could disrupt the association between M18BP1-L and CENP-A nucleosomes by introducing a negative charge near the arginine anchor.

To test the role of S760 in CENP-A nucleosome binding, we mutated serine 760 to alanine (S760A) or to aspartic acid (S760D) to prevent or mimic phosphorylation, respectively. We expressed these proteins and wild-type (WT) M18BP1-L using in vitro transcription and translation. We then assayed the binding of WT M18BP1-L, M18BP1-L[S760A], and M18BP1-L[S760D] to CENP-A nucleosomes by reconstituting CENP-A nucleosomes in vitro on an 18x array of the 601 Widom sequence (Lowary and Widom, 1998) and western blotting the bound protein. WT M18BP1-L and M18BP1-L[S760A] bind to CENP-A nucleosomes in vitro with similar levels, whereas M18BP1-L[S760D] fails to bind to CENP-A nucleosomes (Fig. 1C,D). These results indicate that the S760 phosphorylation site of M18BP1-L regulates CENP-A nucleosome binding in *X. laevis*.

## M18BP1-L phospho-mimetic mutations prevent interphase centromere localization and reduce CENP-A assembly

Next, we tested if phosphorylation of S760 restricts M18BP1 centromere localization to interphase. We assayed centromeric localization of phospho-null M18BP1-L[S760A] and phospho-mimetic M18BP1-L[S760D] on sperm nuclei in interphase *X. laevis* egg extracts. We immunodepleted endogenous M18BP1 (both isoforms) from egg extract and added back in vitro transcribed and translated WT M18BP1-L, M18BP1-L[S760A], or M18BP1-L[S760D] (Fig. 2A,B), and then assayed the localization using immunofluorescence. We found that, consistent with the in vitro nucleosome binding results, M18BP1-L WT and M18BP1-L[S760A] localization to the centromere was normal. In contrast, phospho-mimetic M18BP1-L[S760D] is significantly reduced to ~40% of WT levels (Fig. 2C,D). This is consistent with phosphorylation of M18BP1-L S760 inhibiting centromere localization in *X. laevis* egg extract and CENP-A nucleosome binding in vitro.

Given that the phosphomimetic mutant of M18BP1-L has reduced localization at centromeres during interphase, we tested

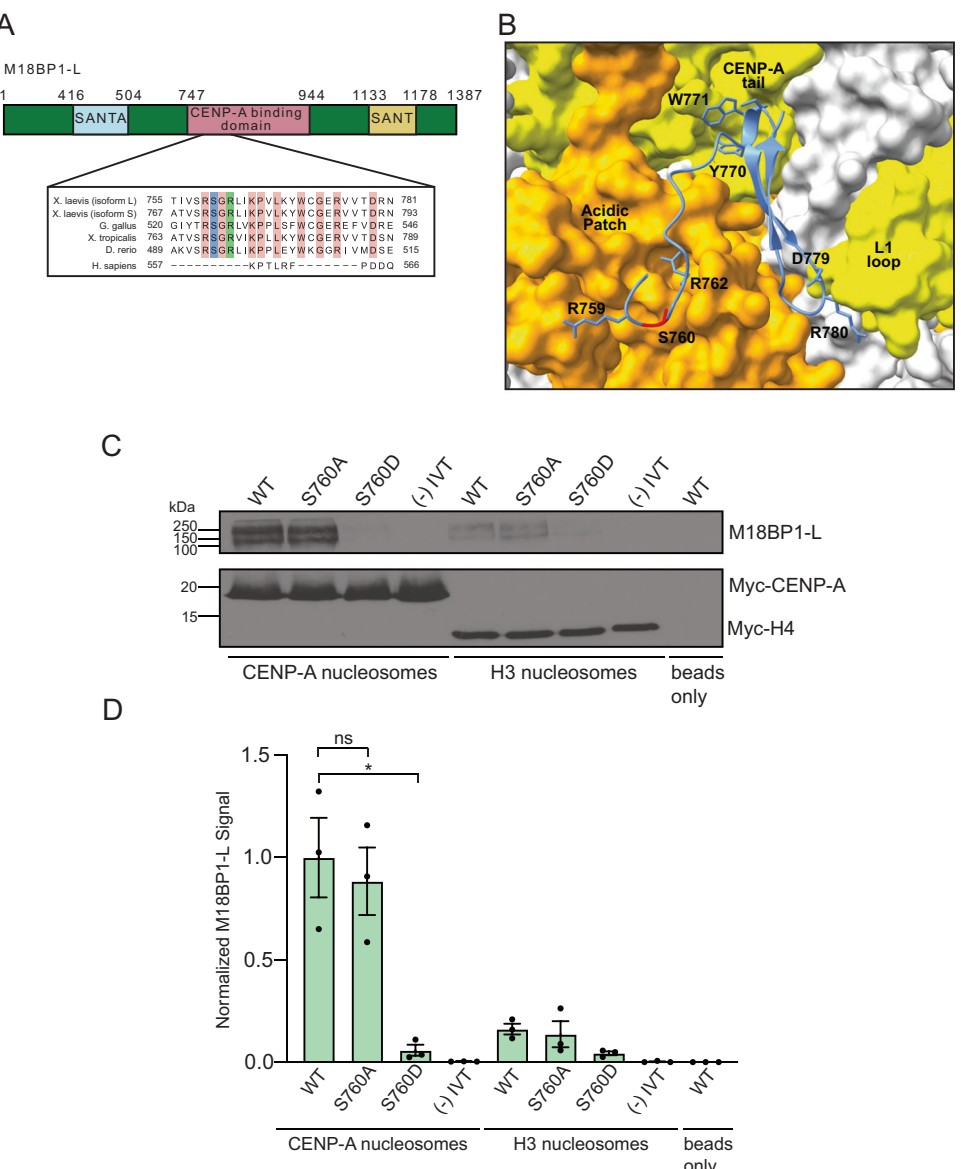

**Figure 1. A conserved phosphorylation site in M18BP1-L regulates binding to CENP-A nucleosomes in vitro.**

(A) Diagram of *X. laevis* M18BP1-L protein. The domain required for CENP-A nucleosome binding is shown in pink, and the SANT and SANTA domains are highlighted in yellow and blue, respectively. The inset shows the region homologous to the CENP-C binding motif, with sequence alignment for *X. laevis* (both isoforms), *G. gallus*, *X. tropicalis*, *D. rerio*, and *H. sapiens*. 100% conserved residues for all species except human are highlighted in pink, the conserved arginine described in (Jiang et al, 2023) is highlighted in green, and the conserved serine shown in this work is highlighted in blue. (B) AlphaFold structural model of the *X. laevis* CENP-A nucleosome bound to M18BP1-L$^{758-789}$ (ipTM & pTM = 0.76) (image modified in ChimeraX). The surface model of the C-term tail and L1 loop of the CENP-A nucleosome is shown in yellow and the surface model of the acidic patch of H2A/H2B is shown in orange. M18BP1-L is shown in blue, with the conserved residues shown to interact with the CENP-A nucleosome in *G. gallus* labeled (Jiang et al, 2023) and residue S760 highlighted in red. (C) Western blot assay of M18BP1 binding to CENP-A or H3 nucleosome arrays. M18BP1 WT, phosphorylation site mutants (S760A and S760D), and a negative control (−) IVT are indicated. The top panel displays an anti-xlM18BP1 western blot, and the bottom panel displays an anti-myc blot to control for chromatin levels. The last lane displays a bead only (no nucleosome) negative control. (D) Quantification of western blots of M18BP1 binding to CENP-A or H3 nucleosomes. The amount of protein bound to chromatin is normalized to the WT M18BP1 binding to CENP-A nucleosomes. Error bars represent SEM of three independent replicates ($n = 3$). *$P = 0.0379$, $^{ns}P = 0.6744$.

whether it affected CENP-A nucleosome assembly in *X. laevis* egg extract. We immunodepleted endogenous M18BP1 from egg extract, then added in vitro transcribed and translated WT M18BP1-L, M18BP1-L$^{S760A}$, or M18BP1-L$^{S760D}$. We supplemented the extract with mRNA encoding V5 epitope-tagged CENP-A (V5-CENP-A) to track new CENP-A assembly via the V5 tag. CENP-A

assembly is lower in M18BP1-L$^{S760D}$, although it is not statistically significantly different from WT M18BP1-L and M18BP1-L$^{S760A}$ ($P = 0.3525$) (Fig. 2E,F). Extracts that were not complemented with M18BP1 (no BP1) showed some CENP-A assembly compared to the control without added V5-CENP-A ((−) IVT), indicating residual assembly activity in the absence of M18BP1 (Fig. 2E,F).

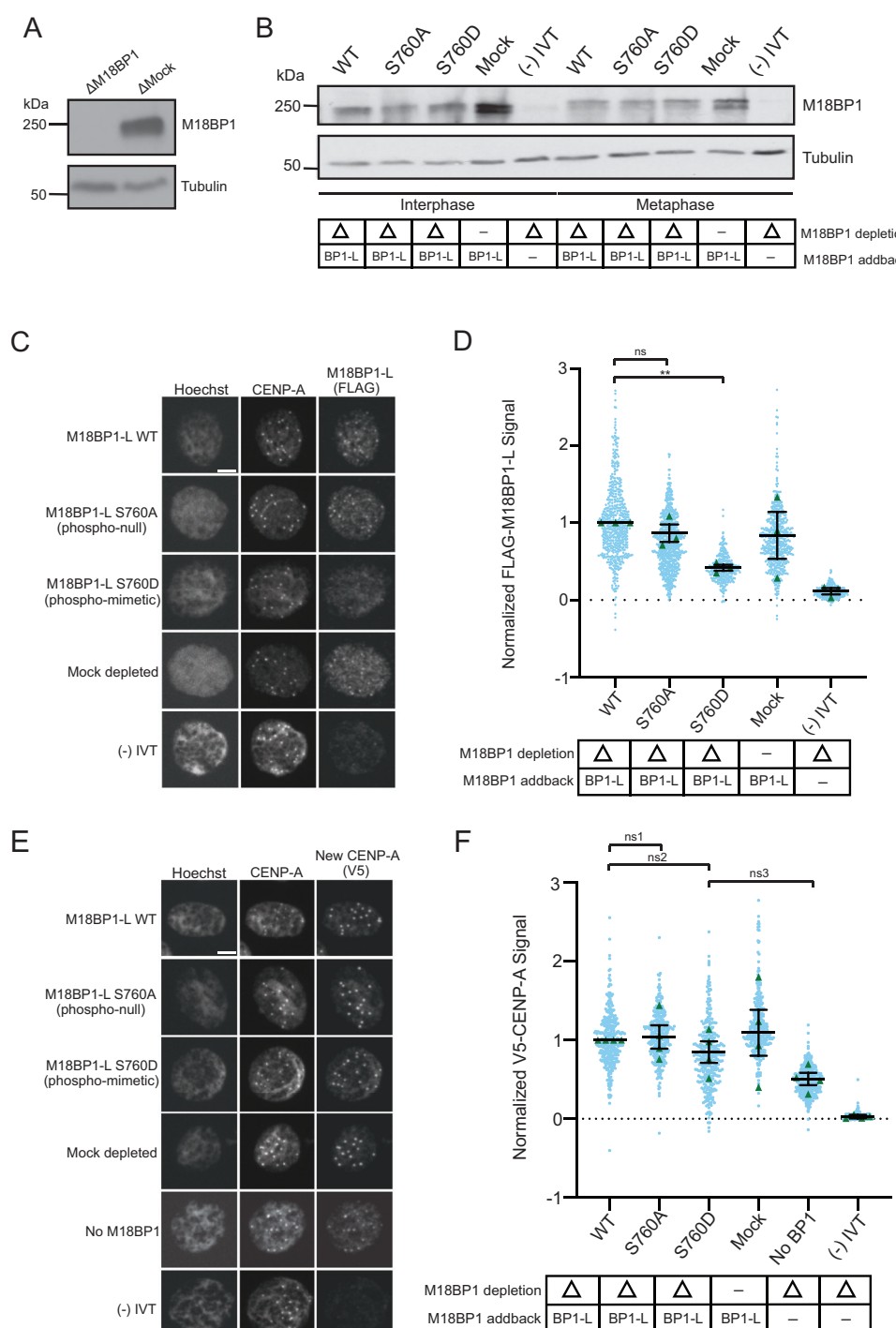

## Removal of CENP-C uncovers a role for M18BP1-L phosphorylation in CENP-A assembly

We previously showed that depletion of CENP-C leads to an increase in M18BP1 localization to the centromere and that full CENP-A assembly requires both CENP-C and M18BP1 (French et al, 2017). Therefore, we tested whether the presence of CENP-C might impact the S760 phosphorylation-dependent localization of M18BP1-L. To do this, we immunodepleted both endogenous M18BP1 and CENP-C and then complemented the extracts with WT or mutant M18BP1-L (Fig. 3A). Consistent with previous results, the levels of WT M18BP1-L at centromeres were higher than the mock depletion condition, in which CENP-C is still present, because CENP-C competes with M18BP1 for CENP-A nucleosome binding (Fig. 3B,C). The phospho-null mutant M18BP1-L[S760A] localized to WT levels. The phospho-mimetic

Figure 2.  Phospho-mimetic mutation of M18BP1-L S760 disrupts centromere localization and inhibits CENP-A assembly.

(A) A representative western blot showing the efficiency of immunodepletion of endogenous M18BP1 from *X. laevis* egg extract. Depletion (Δ) or mock depletion with rabbit IgG is indicated above each column. The M18BP1 blot is shown in the top panel, and a tubulin loading control is shown in the bottom panel. (B) A representative western blot showing the addback of WT or mutant M18BP1 in egg extract. The depletion condition (Δ) is indicated in the table below the blot. The M18BP1 blot is shown in the top panel, and the tubulin loading control is shown in the bottom panel. (C) Representative immunofluorescence images of full-length WT or mutant FLAG-M18BP1-L localization with mock depletion and (−) IVT control (indicated on left) in interphase extract immunodepleted of endogenous M18BP1. Labeling for DNA (Hoechst), total CENP-A, and M18BP1-L (FLAG) is indicated above the image. Scale bar is 5 μm. (D) Quantification of WT or mutant FLAG-M18BP1-L localization with controls (indicated below) in interphase egg extract immunodepleted of endogenous M18BP1. The M18BP1 depletion and addback condition is indicated in the bottom table. The mean signal is normalized to WT FLAG-M18BP1-L localization. Error bars represent SEM of three independent replicates ($n = 3$) with green triangles displaying the mean of each replicate and blue circles representing each individual centromere. $**P = 0.0041$, $^{ns}P = 0.3525$. (E) Representative immunofluorescence images of new V5-CENP-A assembly in interphase extract immunodepleted of endogenous M18BP1, then supplemented with full-length WT or mutant FLAG-M18BP1-L, a mock depletion with rabbit IgG, a (−) IVT control without V5-CENP-A, or no M18BP1 addback (no BP1) (indicated on left). Labeling for DNA (Hoechst), total CENP-A, and new CENP-A (V5) is indicated above the image. Scale bar is 5 μm. (F) Quantification of new V5-CENP-A with controls (indicated below) in interphase egg extract immunodepleted of endogenous M18BP1. M18BP1 depletion and addback condition are indicated in the bottom table. The signal is normalized to the WT FLAG-M18BP1-L addback condition. Error bars represent SEM of four independent replicates ($n = 4$) with green triangles displaying the mean of each replicate and blue circles representing each individual centromere. $^{ns1}P = 0.8053$, $^{ns2}P = 0.3465$, $^{ns3}P = 0.0821$.

M18BP1-L$^{S760D}$ had reduced localization in the absence of CENP-C, to ~30% WT levels. This shows that the phospho-mimetic S760D inhibits the binding of M18BP1-L in the absence of CENP-C, despite the availability of more CENP-A nucleosome binding sites (Fig. 3B,C). Thus, in the presence or absence of CENP-C, phosphomimetic M18BP1-L$^{S760D}$ is reduced in its levels at centromeres.

Because CENP-C and M18BP1 are required for CENP-A assembly in *X. laevis* (French et al, 2017), we tested whether the absence of CENP-C would exacerbate the effect of M18BP1-L S760 phosphorylation on CENP-A assembly. We repeated the experiment with immunodepletion of both endogenous M18BP1 and CENP-C from egg extract, followed by complementation with M18BP1-L and V5-tagged CENP-A. In the absence of CENP-C, the addition of M18BP1-L$^{S760D}$ causes a reduction in new CENP-A assembly compared to the addition of WT M18BP1-L, M18BP1-L$^{S760A}$, or mock-depleted extract (Fig. 3D,E). These data show that phosphorylation of M18BP1-L controls its proper localization to centromeres and CENP-A assembly, and that both M18BP1-L and CENP-C contribute to proper CENP-A assembly.

## Dephosphorylation of M18BP1-L is not sufficient for bypass of metaphase inhibition of centromere localization or CENP-A assembly

M18BP1-L does not localize to the centromere in metaphase, unlike its isoform M18BP1-S, which localizes by binding to CENP-C (French and Straight, 2019; Flores Servin et al, 2023). We tested whether lack of S760 phosphorylation would be sufficient to cause metaphase centromere localization and CENP-A assembly by immunodepleting endogenous M18BP1 from metaphase extracts, complementing the extract with WT M18BP1-L, M18BP1-L$^{S760A}$, or M18BP1-L$^{S760D}$, and then assaying M18BP1-L localization to sperm nuclei. WT M18BP1-L, M18BP1-L$^{S760A}$, and M18BP1-L$^{S760D}$ all failed to localize to the centromere in metaphase egg extract (Fig. 4A,B). M18BP1 binds to the CENP-A nucleosome through a conserved CENP-C motif (Carroll et al, 2010; Kato et al, 2013; Ariyoshi et al, 2021; Guo et al, 2017). Because CENP-C competes for the M18BP1 binding site on the CENP-A nucleosome and localizes to the centromere in metaphase, we tested whether CENP-C could compete with M18BP1-L$^{S760A}$ for metaphase CENP-A nucleosome binding. We immunodepleted both CENP-C and M18BP1, complemented the extract with WT M18BP1-L, M18BP1-

L$^{S760A}$, and M18BP1-L$^{S760D}$, and assayed M18BP1-L localization. Similar to conditions where CENP-C is present, in the absence of CENP-C, M18BP1-L fails to localize to centromeres regardless of S760 mutation (Fig. 4C,D). In addition, CENP-A assembly does not occur in metaphase in all phospho-mutant and immunodepletion conditions (Fig. EV3A–D). This indicates that additional activities in addition to inhibitory phosphorylation prevent M18BP1-L binding to CENP-A nucleosomes and CENP-A assembly in metaphase.

## CENP-N does not prevent M18BP1-L binding to metaphase centromeres

The only other protein known to bind to CENP-A nucleosomes is CENP-N (Carroll et al, 2010, 2009). CENP-N interacts with the L1 loop and DNA of CENP-A nucleosomes (Pentakota et al, 2017; Carroll et al, 2009; Fang et al, 2015; Guo et al, 2017; Carroll et al, 2010; Chittori et al, 2018). When incorporated into the CCAN, CENP-N is unable to directly contact the CENP-A nucleosome (Pesenti et al, 2022; Yatskevich et al, 2022; McKinley et al, 2015; Allu et al, 2019). Thus, an interesting possibility is that CENP-N might bind to CENP-A nucleosomes independent of the CCAN in metaphase. In *G. gallus*, CENP-N and CENP-C compete for binding to CENP-A nucleosomes in vitro, and it was proposed that CENP-C increases its affinity for CENP-A nucleosomes in metaphase due to phosphorylation and excludes CENP-N (Ariyoshi et al, 2021). The central domain of *X. laevis* CENP-C lacks the phosphorylation site conserved in humans and *G. gallus* that promotes CENP-A nucleosome binding but retains a threonine at the equivalent residue to S760 in M18BP1. Thus, we tested whether a change in the mode of CENP-C binding might allow CENP-N to engage CENP-A nucleosomes in metaphase and prevent M18BP1-L binding by competing for binding to CENP-A. We immunodepleted endogenous CENP-C from interphase and metaphase egg extracts and assayed endogenous CENP-N localization (Fig. 5A). In the absence of CENP-C, CENP-N does not localize to metaphase centromeres (Fig. 5B,D). Thus, it is not possible for CENP-N to block M18BP1-L localization to metaphase centromeres in the absence of CENP-C. Interestingly, CENP-A levels are affected by the depletion of CENP-C. The overall levels of CENP-A at metaphase centromeres significantly decrease with the loss of CENP-C. On the other hand, the interphase CENP-A levels are not significantly affected by CENP-C loss (Fig. 5B,C). This

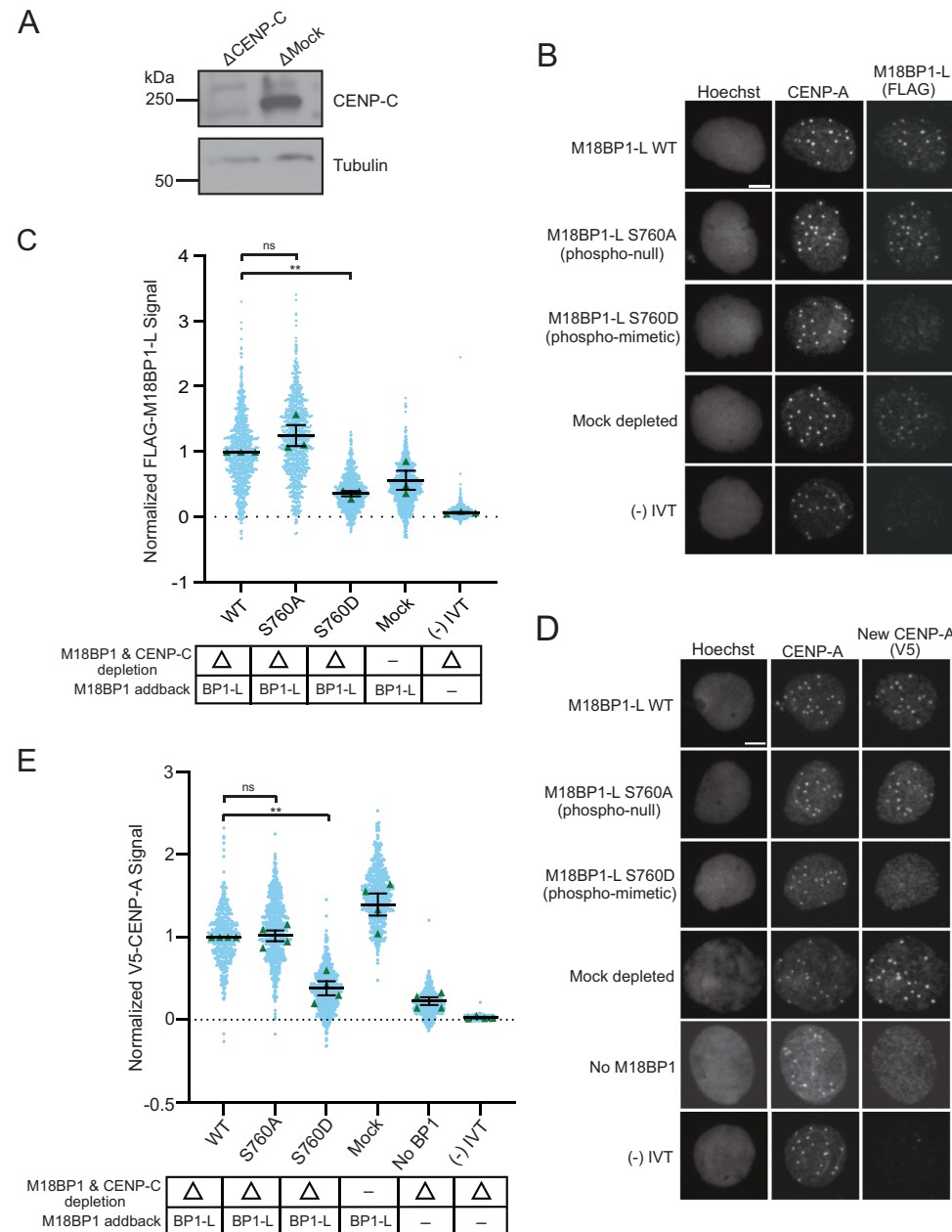

**Figure 3. Removal of CENP-C exacerbates defects in CENP-A localization and CENP-A assembly of phospho-mimetic M18BP1^S760D.**

(A) A representative western blot displaying immunodepletion of endogenous CENP-C and an IgG mock depletion in egg extract. The depletion condition (Δ) is indicated above each column. The CENP-C blot is shown in the top panel, and a tubulin loading control is shown in the bottom panel. (B) Representative immunofluorescence images of full-length WT or mutant FLAG-M18BP1-L localization with mock depletion and (−) IVT control (indicated on left) in interphase extract immunodepleted of endogenous CENP-C and M18BP1. Labeling for DNA (Hoechst), total CENP-A, and M18BP1-L (FLAG) is indicated above the image. Scale bar is 5 μm. (C) Quantification of WT or mutant FLAG-M18BP1-L localization with controls (indicated below) in interphase egg extract immunodepleted of endogenous CENP-C and M18BP1. CENP-C and M18BP1 depletion and addback conditions are indicated in the bottom table. The signal is normalized to WT FLAG-M18BP1-L localization. Error bars represent SEM of three independent replicates ($n = 3$) with green triangles displaying the mean of each replicate and blue circles representing each individual centromere. **$P = 0.0041$, $^{ns}P = 0.2570$. (D) Representative immunofluorescence images of new V5-CENP-A assembly in interphase extract immunodepleted of endogenous CENP-C and M18BP1, then supplemented with full-length WT or mutant FLAG-M18BP1-L, a mock depletion, a (−) IVT control without V5-CENP-A, or no M18BP1 addback (no BP1) (indicated on left). Labeling for DNA (Hoechst), total CENP-A, and new CENP-A (V5) is indicated above the image. Scale bar is 5 μm. (E) Quantification of new V5-CENP-A with controls (indicated below) in interphase egg extract immunodepleted of endogenous CENP-C and M18BP1. CENP-C and M18BP1 depletion and addback conditions are indicated in the bottom table. The signal is normalized to the WT FLAG-M18BP1-L addback condition. Error bars represent SEM of four independent replicates ($n = 4$) with green triangles displaying the mean of each replicate and blue circles representing each individual centromere. **$P = 0.0056$, $^{ns}P = 0.8257$.

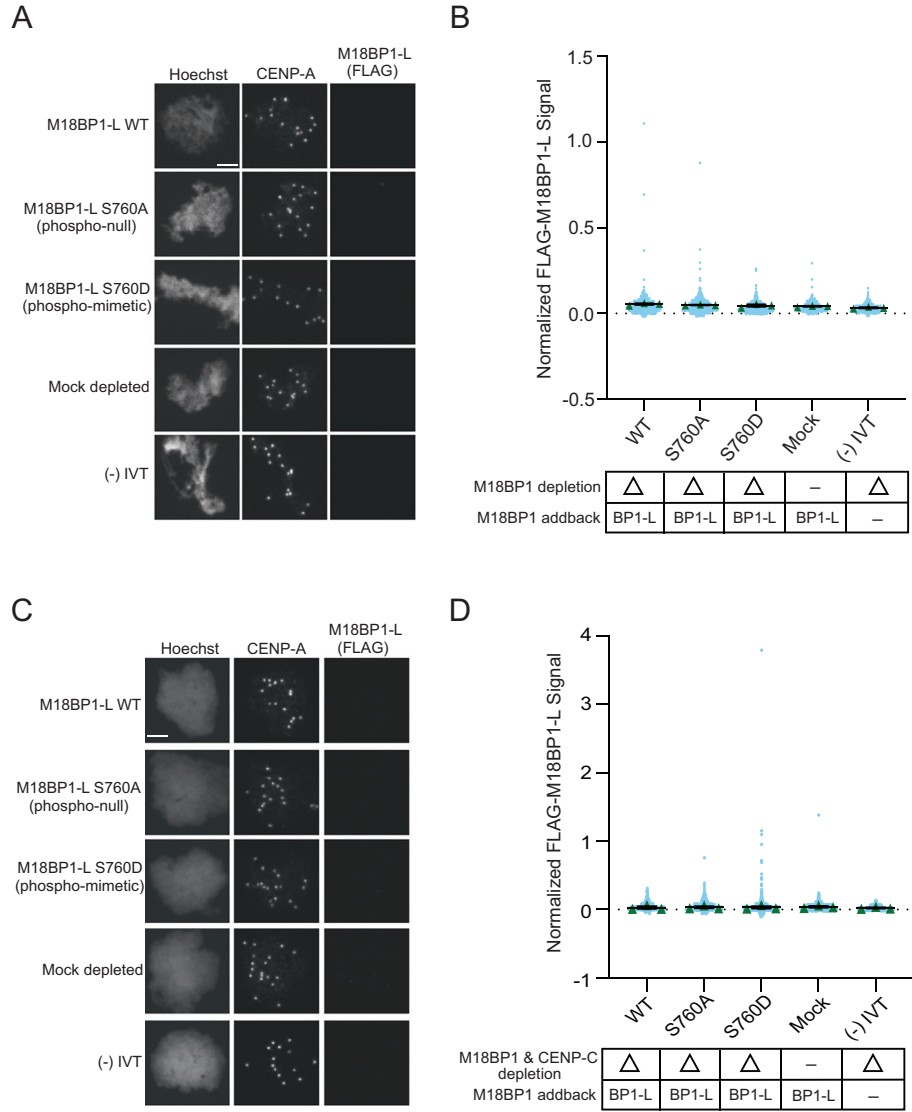

**Figure 4. The non phosphorylatable mutant M18BP1[S760A] does not bypass the metaphase inhibition of M18BP1 localization.**

(**A**) Representative immunofluorescence images of full-length WT or mutant FLAG-M18BP1-L localization with mock depletion and (−) IVT control (indicated on left) in metaphase extract immunodepleted of endogenous M18BP1. Labeling for DNA (Hoechst), total CENP-A, and M18BP1-L (FLAG) is indicated above the image. Scale bar is 5 μm. (**B**) Quantification of WT or mutant FLAG-M18BP1-L localization with controls (indicated below) in metaphase egg extract immunodepleted of endogenous M18BP1. The M18BP1 depletion and addback condition is indicated in the bottom table. The signal is normalized to WT FLAG-M18BP1-L localization. Error bars represent SEM of three independent replicates ($n = 3$) with green triangles displaying the mean of each replicate and blue circles representing each individual centromere. (**C**) Representative immunofluorescence images of full-length WT or mutant FLAG-M18BP1-L localization with mock depletion and (−) IVT control (indicated on left) in metaphase extract immunodepleted of endogenous CENP-C and M18BP1. Labeling for DNA (Hoechst), total CENP-A, and M18BP1-L (FLAG) is indicated above the image. Scale bar is 5 μm. (**D**) Quantification of WT or mutant FLAG-M18BP1-L localization with controls (indicated below) in metaphase egg extract immunodepleted of endogenous CENP-C and M18BP1. The CENP-C and M18BP1 depletion and addback condition is indicated in the bottom table. The signal is normalized to WT FLAG-M18BP1-L localization. Error bars represent SEM of three independent replicates ($n = 3$) with green triangles displaying the mean of each replicate and blue circles representing each individual centromere.

indicates that CENP-C could be playing a role in the retention and stability of CENP-A nucleosomes at the metaphase centromere, because CSF (cytostatic factor)-arrested (metaphase) egg extract has not gone through the CENP-A assembly process, and thus the reduction in CENP-A cannot be due to a loss of CENP-A assembly. In addition, we found that CENP-N still localizes to the centromere in interphase in the absence of CENP-C, albeit at low levels (Fig. 5B,D). This supports the possibility that CENP-N binds to CENP-A nucleosomes independent of the CCAN in interphase.

## Discussion

CENP-A nucleosome assembly is decoupled from DNA replication in vertebrates, unlike the assembly of most replication-coupled H3 nucleosomes, and instead occurs in G1/interphase (Jansen et al, 2007; Maddox et al, 2007; Moree et al, 2011; Bernad et al, 2011; Silva et al, 2012). In this work, we uncover a new mechanism, wherein the phosphorylation of the conserved site S760 in *X. laevis* M18BP1-L, located in the CENP-C motif, regulates the binding of

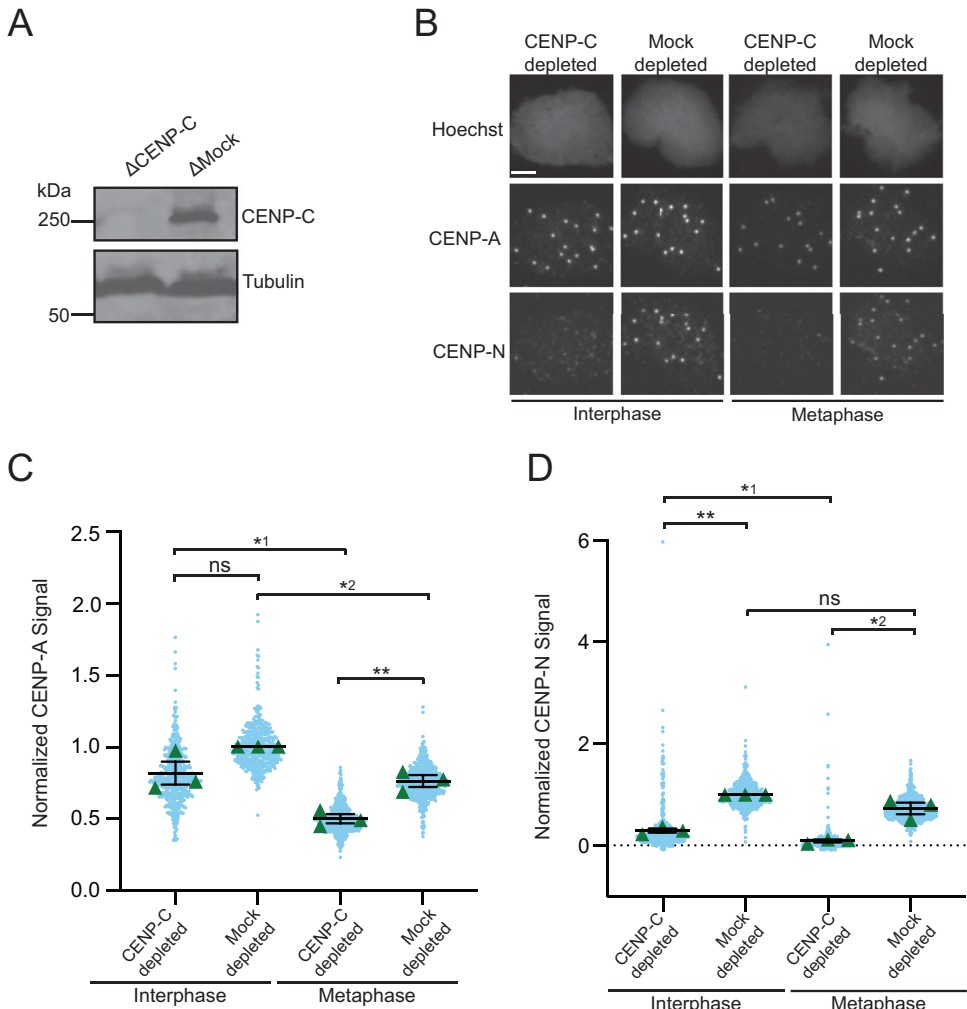

**Figure 5. Centromeric localization of CENP-N in the absence of CENP-C.**

(A) A representative western blot displaying immunodepletion of endogenous CENP-C and an IgG mock depletion in egg extract. The depletion condition (Δ) is indicated above each column. The CENP-C blot is shown in the top panel, and a tubulin loading control is shown in the bottom panel. (B) Representative immunofluorescence images of endogenous CENP-N localization with mock depletion or immunodepletion of endogenous CENP-C in interphase or metaphase egg extract. Labeling for DNA (Hoechst), total CENP-A, and CENP-N is indicated above the image. Scale bar is 5 μm. (C) Quantification of all CENP-A signal in interphase or metaphase egg extract immunodepleted of endogenous CENP-C or mock-depleted. The CENP-C depletion condition is indicated in the bottom table. The signal is normalized to interphase, mock-depleted CENP-A signal. Error bars represent SEM of three independent replicates ($n = 3$) with green triangles displaying the mean of each replicate and blue circles representing each individual centromere. $**P = 0.0081$, $*^{1}P = 0.0435$, $*^{2}P = 0.0277$, $^{ns}P = 0.1467$. (D) Quantification of endogenous CENP-N localization in interphase or metaphase egg extract immunodepleted of endogenous CENP-C or mock-depleted. The CENP-C depletion condition is indicated in the bottom table. The signal is normalized to the CENP-A signal within each condition to account for the variance in CENP-A levels, and then further normalized to interphase, mock-depleted CENP-N localization. Error bars represent SEM of three independent replicates ($n = 3$) with green triangles displaying the mean of each replicate and blue circles representing each individual centromere. $**P = 0.0034$, $*^{1}P = 0.0190$, $*^{2}P = 0.0253$, $^{ns}P = 0.1390$.

M18BP1-L to CENP-A nucleosomes and the assembly of new CENP-A nucleosomes in interphase (Fig. 6). The mechanism for phosphorylation disrupting nucleosome binding is likely through the introduction of the negatively charged phosphate group proximal to the positive arginine anchor that engages the acidic patch on H2A/H2B of the nucleosome. This phospho-regulated binding of M18BP1 to CENP-A nucleosomes is a new mechanism that controls the timing of CENP-A assembly.

We tested whether preventing phosphorylation of S760 might be sufficient to drive M18BP1-L binding to CENP-A nucleosomes in metaphase by complementing metaphase extract with S760A mutant M18BP1-L. We showed that the M18BP1-L S760A mutation was unable to bypass the metaphase inhibition of centromere localization, despite our demonstration that M18BP1-L^S760A could bind to CENP-A nucleosomes in vitro. This indicates that there is another regulatory mechanism preventing M18BP1-L from binding to CENP-A in metaphase. Other post-translational modifications might alter M18BP1-L binding to CENP-A nucleosomes in metaphase. However, we did not detect additional phosphorylation near the regions of M18BP1-L or ggKNL2 that interact with CENP-A (Fig. EV1B) (French et al, 2017).

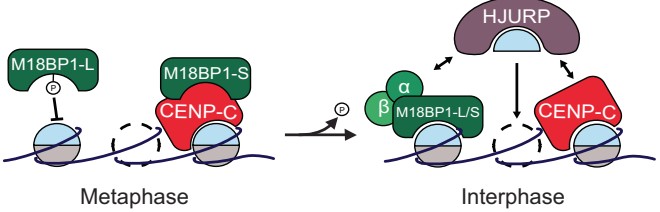

Metaphase · Interphase

**Figure 6. Model of the phospho-regulation of M18BP1-L.**

Model showing phosphorylation-dependent inhibition of M18BP1-L binding to CENP-A nucleosomes and M18BP1-S binding to CENP-C in metaphase (left). Dephosphorylation triggers a switch to CENP-A nucleosome binding and new CENP-A assembly in interphase (right). In interphase, both the Mis18 complex and CENP-C bind to HJURP to recruit new CENP-A for assembly.

An alternative mechanism for preventing M18BP1-L binding to CENP-A in metaphase is that another protein regulates the interaction by binding the CENP-A nucleosome or by binding M18BP1-L. We show that neither CENP-C nor CENP-N is responsible for inhibiting M18BP1-L binding to centromeres in metaphase, thus if another protein is preventing M18BP1-L centromere localization, it remains to be identified. Interestingly, we found that CENP-N localizes to the centromere independent of CENP-C during interphase. Because CENP-C is required for CCAN formation (Yatskevich et al, 2022; Pesenti et al, 2022), this indicates that CENP-N can bind to CENP-A nucleosomes independent of the CCAN during interphase. Whether CENP-N binds to the interphase centromere through the CCAN, through direct CENP-A nucleosome interaction, or both remains unclear. It will be interesting to determine whether CENP-N binds to the centromere through different binding modes during the cell cycle and whether different modes of engagement regulate distinct processes, including CCAN and kinetochore formation, CENP-A assembly, and centromere structure (Zhou et al, 2022; Carroll et al, 2009; Pentakota et al, 2017; Chittori et al, 2018).

Interestingly, we found that overall CENP-A levels decrease with CENP-C depletion, with the most significant reduction found in metaphase. A slight reduction in CENP-A levels in interphase can be explained by the requirement of CENP-C for new CENP-A assembly, as a loss of new assembly will lead to an overall reduction in CENP-A. However, the reduction in CENP-A in CSF-arrested (metaphase) egg extract supplemented with sperm chromatin will be unaffected by CENP-A assembly, as it has yet to occur. This suggests that previously existing CENP-A is being lost, indicating that CENP-C and/or CENP-N may be stabilizing CENP-A nucleosomes at the centromere in *X. laevis*. CENP-C and/or CENP-N stabilization of pre-existing CENP-A nucleosomes has been shown both in vitro and in humans, although there is conflicting evidence regarding the exact degree of stability it grants in vivo (Falk et al, 2015; Cao et al, 2018; Guo et al, 2017). The dependence of CENP-A nucleosome stability on CENP-C and/or CENP-N has not yet been assayed in *X. laevis*, and this data suggests that a similar mechanism may exist, although further studies are required to directly test it.

The identity of the kinase that phosphorylates M18BP1-L S760 is unclear. Cdk1 and Plk1 phosphorylate M18BP1 in vertebrates, but S760 is not located within a consensus motif for either kinase

(McKinley and Cheeseman, 2014; Pan et al, 2017; French and Straight, 2019; Stankovic et al, 2017; Spiller et al, 2017; Conti et al, 2024; Parashara et al, 2024). The consensus sites for other mitotic kinases, including AuroraA, AuroraB (Cheeseman et al, 2002; Sardon et al, 2010), Mps1 (von Schubert et al, 2015), Bub1 (Elowe and Bolanos-Garcia, 2022), Haspin (Maiolica et al, 2014), and Nek2A (Fry, 2002; Fry et al, 1995) are also not well matched to the sequence surrounding the S760 residue.

The phosphoregulation of M18BP1-S binding to CENP-A nucleosomes also remains an open question. We were unable to detect any peptides containing the conserved S772 residue in M18BP1-S with mass spectrometry (Fig. EV1C). M18BP1-S binds directly to CENP-C in metaphase to localize to centromeres; thus, it will be interesting to determine if M18BP1-S is phosphorylated to release it from CENP-A nucleosomes and enable its binding to CENP-C. While full-length M18BP1-L does not localize to the metaphase centromere, the fragment M18BP1-L$^{161-570}$ binds CENP-C in the same way as full-length M18BP1-S (French and Straight, 2019). This indicates that the C-terminus of M18BP1-L either prevents binding to CENP-C or inhibits centromere localization through another mechanism. A recent report showing that artificial dimerization of human M18BP1 is sufficient for centromere targeting, independent of Mis18α/β, raises the possibility that *X. laevis* M18BP1 must dimerize to bind to CENP-C during mitosis, and that the C-terminus of M18BP1-L might prevent this dimerization (Walstein et al, 2025).

The mechanisms that couple CENP-A assembly to mitotic exit involve a series of phosphorylation-dependent protein-protein interactions that inhibit CENP-A assembly until cells complete chromosome segregation and enter G1/interphase. These include multiple steps that alter the activities of the Mis18 complex and CENP-C, the primary adapters for HJURP and soluble CENP-A recruitment to centromeres. We have discovered an additional control mechanism that prevents CENP-A assembly in metaphase by phosphorylating M18BP1 to prevent the interaction of the CENP-C motif in M18BP1 with the acidic patch of the nucleosome. M18BP1 binds to CENP-A nucleosomes using the C-terminal tail and L1 loop which provide specificity for CENP-A and the acidic patch which increases the affinity of the protein for the nucleosome. Human M18BP1 doesn't directly bind to the CENP-A nucleosome and also lacks the conserved serine residue that we identified in this work. However, in the CENP-C protein, the same residue is conserved in the CENP-C motifs across humans, *X. laevis*, and *G. gallus*. It will be interesting to determine whether CENP-C interaction might also be regulated by phosphorylation near the arginine anchor and whether this represents a more generalizable mechanism for tuning the affinity of proteins that bind to centromeric or other nucleosomes.

## Methods

### Reagents and tools table

| Reagent/resource | Reference or source | Identifier or catalog number |
|---|---|---|
| **Experimental models** | | |
| *Xenopus laevis* females | Nasco | #LM00535MX |
| **Recombinant DNA** | | |
| V5-xlCENP-A in pCS2+ | Aaron Straight, Stanford University | ASP3631 |

| Reagent/resource | Reference or source | Identifier or catalog number |
|---|---|---|
| xlHJURP in pCS2+ | Aaron Straight, Stanford University | ASP2260 |
| xlM18BP1-L-3xFLAG in pCS2+ | Aaron Straight, Stanford University | ASP1871 |
| xlM18BP1-L$^{S760A}$-3xFLAG in pCS2+ | Aaron Straight, Stanford University | ASP4375 |
| xlM18BP1-L$^{S760D}$-3xFLAG in pCS2+ | Aaron Straight, Stanford University | ASP4376 |
| Myc-xlCENP-A and H4 in pST39 | Aaron Straight, Stanford University | ASP3451 |
| 18 × 601 array DNA in pUC18 | Aaron Straight, Stanford University | ASP696 |
| H2A in pET3a | Aaron Straight, Stanford University | ASP1120 |
| H2B in pET3a | Aaron Straight, Stanford University | ASP1121 |
| H3 in pET3A | Aaron Straight, Stanford University | ASP1122 |
| Myc-H4 in pST39 | Aaron Straight, Stanford University | ASP2701 |
| MBP-M18BP1-L$^{747-944}$ in pMalc2x | Aaron Straight, Stanford University | ASP3194 |
| MBP-M18BP1-S$^{750-917}$ in pMalc2x | Aaron Straight, Stanford University | ASP3192 |
| **Antibodies** | | |
| Rabbit anti-xlCENP-C | Aaron Straight, Stanford University | Raised and purified against xlCENP-C$^{207-296}$ (Milks et al, 2009) |
| Rabbit anti-xlM18BP1 | Aaron Straight, Stanford University | Raised against GST-xlM18BP1-L$^{161-415}$ and purified against xlM18BP1.S$^{161-375}$ (Moree et al, 2011) |
| whole rabbit IgG | Jackson ImmunoResearch Laboratories, Inc. | 011-000-003 |
| Rabbit anti-xlCENP-A | Aaron Straight, Stanford University | Raised and purified against xlCENP-A$^{1-50}$ (Milks et al, 2009) |
| Mouse anti-FLAG | Millipore Sigma | F1804 |
| Rabbit anti-xlCENP-N | Aaron Straight, Stanford University | Raised and purified against xlCENP-N peptide acetyl-CPHKARNSFKITEKR-amide (Moree et al, 2011) |
| Mouse anti-V5 | ThermoFisher | R960-25 |
| Alexa 647 goat anti-mouse IgG | Invitrogen | A21236 |
| Alexa 568 goat anti-rabbit IgG | Invitrogen | A11011 |
| Alexa 488 goat anti-rabbit IgG | Invitrogen | A11034 |
| Mouse anti-DM1α (tubulin) | Sigma | T6199 |
| Mouse anti-myc clone 4A6 | Sigma | 05-724MG |
| Horseradish peroxidase (HRP)-conjugated goat anti-mouse secondary | Bio-Rad | 1706516 |
| Horseradish peroxidase (HRP)-conjugated goat anti-rabbit secondary | Bio-Rad | 1706515 |
| Rabbit anti-MBP | Aaron Straight, Stanford University | NA |
| **Oligonucleotides and other sequence-based reagents** | | |
| α-thio-dCTP | ChemCyte | CC-3002-1 |
| α-thio-dGTP | ChemCyte | CC-3003-1 |
| α-thio-dTTP | ChemCyte | CC-3004-1 |
| biotin-14-dATP | Invitrogen | 19524016 |

| Reagent/resource | Reference or source | Identifier or catalog number |
|---|---|---|
| Reverse primer for generating xlM18BP1-L S760 point mutation | ATGGTATCATTAGACAGCGGTGATG | ASO5688 |
| Forward primer for generating xlM18BP1-L S760A point mutation | CGTGAGCCGCGCTGGCCGACTCA | ASO5686 |
| Forward primer for generating xlM18BP1-L S760D point mutation | cgtgagccgcgatggccgactca | ASO5687 |
| **Chemicals, enzymes, and other reagents** | | |
| Pregnant Mare Serum Gonadotropin (PMSG) | BioVendor LLC | RP1782725000 |
| Human Chorionic Gonadotropin (HCG) | Merck Chorulon | CH-475-1 |
| Roche cOmplete™ EDTA-free protease inhibitor cocktail tablet | Millipore Sigma | 11873580001 |
| Klenow fragment 3'-5' exo- | New England BioLabs | M0212L |
| Streptavidin-Alexa 647 | Invitrogen | S32357 |
| SyBr Gold | Life Technologies | S11494 |
| streptavidin-coated M-280 Dynabeads | Invitrogen | 11205D |
| Promega SP6 TNT Quick Coupled Transcription/Translation System | Promega | L2080 |
| Protein A Dynabeads | Invitrogen | 10001D |
| mMessage mMachine SP6 Transcription kit | Fisher Scientific | AM1340 |
| Alexa Fluor 594 | Invitrogen | A10239 |
| Hoechst | ThermoFisher | H3570 |
| Nitrocellulose Membrane | Amersham | GE10-6000-00 |
| Pierce ECL Western Blotting Substrate | ThermoFisher | 32106 |
| GelCode Blue Stain Reagent | ThermoFisher | 24590 |
| **Software** | | |
| Mass_Spec | Aaron Straight, Stanford University (French et al, 2017) | https://gitlab.com/straightlab/mass_spec |
| PSI-BLAST | NCBI | https://blast.ncbi.nlm.nih.gov/Blast.cgi |
| MAFFT | Katoh et al, 2002 | https://mafft.cbrc.jp/alignment/software/ |
| ImageAnalysisTools (v5.1.5) | Aaron Straight, Stanford University (Moree et al, 2011) | https://github.com/cjfuller/imageanalysistools |
| ImageJ v2.16.0/1.54p; FIJI | NIH | https://fiji.sc/ |
| Prism | GraphPad | https://www.graphpad.com/ |
| AlphaFold | EMBL-EBI | https://alphafold.com/ |
| ChimeraX | UCSF | https://www.cgl.ucsf.edu/chimerax/ |
| **Other** | | |
| EmulsiFlex-C5 | Avestin, Inc. | NA |
| Sonifier 250 | Branson | 100-132-135 |
| HiTrap Q HP column | Cytiva | 17115401 |

| Reagent/resource | Reference or source | Identifier or catalog number |
|---|---|---|
| HiTrap SP HP column | Cytiva | 17115201 |
| HiLoad 16/600 Superdex 200 column | Cytiva | 28989335 |
| Beckman 45Ti rotor | Beckman Coulter | 339160 |
| Hydroxyapatite Column | Bio-Rad | 1572000 |
| Amicon Ultra Centrifugal Filter | Millipore Sigma | UFC801008 |
| Gigaprep kit | Qiagen | 12191 |
| 13 × 51 mm polyallomer tube | Beckman Coulter | 326819 |
| International Clinical Centrifuge | IEC | model CL no. 740250; rotor model 221 |
| SW55Ti Rotor | Beckman Coulter | 342196 |
| RNeasy mini columns | Qiagen | 74104 |
| Glass Centrifuge Tubes | Kimble | 45500-15 |
| Nikon Eclipse Ti2 inverted microscope | Nikon | TI2-LA-FL-2 |
| CrestOptic X-Light V3 Confocal Unit | Nikon | NA |
| Celesta Light Engine | Lumencor | NA |
| EMCCD Camera | iXon Life | 897 |
| SRX-101A Film Processor | Konica | NA |

## Experimental model details

*Xenopus laevis* females (Nasco catalog #LM00535MX) were housed and maintained in the Stanford Veterinary Service Center. Frogs were primed 2–14 days prior to ovulation by subcutaneous injection of 50U of Pregnant Mare Serum Gonadotropin (PMSG, BioVendor LLC RP1782725000) in the dorsal lymph sac. Ovulation was induced for 16–18 h via subcutaneous injection of 500U Human Chorionic Gonadotropin (HCG, Merck Chorulon Order Code No. CH-475-1) in the dorsal lymph sac. During ovulation, frogs were housed individually in 2 L of 1×MMR buffer (6 mM Na-HEPES pH 7.8, 0.1 mM EDTA, 100 mM NaCl, 2 mM KCl, 1 mM $MgCl_2$, 2 mM $CaCl_2$) at 17 °C. All animal work was carried out following the guidelines of the Stanford University Administrative Panel on Laboratory Animal Care (APLAC) under protocol number 9739.

## Protein purification

H2A/H2B dimers and H3/H4 tetramers were purified as described in Guse et al, 2012, and Westhorpe et al, 2015. Histones H2A, H2B, H3, and H4 were expressed from a pST39 vector in BL21-Codon Plus (DE3)-RIPL *Escherichia coli* competent cells and grown in 2xYT media (20 g/L tryptone, 10 g/L yeast extract, 5 g/L NaCl) at 37 °C to $OD_{600}$ of 0.6, then induced with 0.25 mM isopropyl β-D-1-thiogalactopyranoside (IPTG) for 3 h at 37 °C. The bacterial cultures were pelleted and resuspended in lysis buffer (20 mM Potassium Phosphate ($KPO_4$) pH 6.8, 1 M NaCl, 5 mM β-mercaptoethanol (β-me), 1 mM phenylmethylsulfonyl fluoride (PMSF), 1 mM benzamidine, 0.05% NP-40, and 0.2 mg/mL lysozyme). The sample was lysed using an EmulsiFlex-C5 (Avestin, Inc.) and 1 × 30 s sonication at duty cycle 50% and output 8 on a Branson Sonicator. The lysed sample was

centrifuged at 4 °C at $18,000 \times g$ for 20 min, and the insoluble pellet was washed in lysis buffer, then resuspended in unfolding buffer (7 M guanidine-HCl, 20 mM Tris-HCl pH 7.5, 10 mM DTT). The sample was recentrifuged at 4 °C at $18,000 \times g$ for 20 min, and the supernatant was dialyzed into urea buffer (6 M deionized urea, 200 mM NaCl, 10 mM Tris-HCl pH 8, 1 mM EDTA, 5 mM β-me, 0.1 mM PMSF). The dialyzed supernatant was then loaded onto 5 mL HiTrap Q HP and HiTrap SP HP columns (Cytiva 17115401 and Cytiva 17115201, respectively) in series. Histones were eluted from the Histrap SP column with urea buffer containing 1 M NaCl, dialyzed into water, and lyophilized. To reconstitute the H2A/H2B dimer and H3/H4 tetramer, histones were mixed in an equimolar ratio and resuspended in unfolding buffer. They were then dialyzed into 2 M NaCl, 10 mM Tris-HCl pH 7.6, 1 mM EDTA, and 5 mM β-me. The samples were then loaded onto a HiLoad 16/600 Superdex 200 column (Cytiva 28989335), and the H2A/H2B dimer and H3/H4 tetramer were purified by size-exclusion chromatography.

*X. laevis* CENP-A-myc/H4 tetramers were expressed and purified as outlined in (Flores Servin et al, 2023) and adapted from (Guse et al, 2012). CENP-A-myc/H4 was expressed from a pST39 vector in BL21-Codon Plus (DE3)-RIPL *Escherichia coli* competent cells and grown in 2xYT media (20 g/L tryptone, 10 g/L yeast extract, 5 g/L NaCl) at 37 °C until OD600 was 0.3–0.4, then moved to 20 °C and grown until $OD_{600}$ was 0.5–0.6. Cultures were then induced for 4 h with 0.2 mM IPTG at 20 °C. The bacterial cultures were pelleted and resuspended in lysis buffer (20 mM $KPO_4$, 1 M NaCl, 5 mM β-me, and 1 Roche cOmplete™ EDTA-free protease inhibitor cocktail tablet (Millipore Sigma 11873580001)). The sample was homogenized using an EmulsiFlex-C5 (3 rounds of 10,000–15,000 psi) (Avestin, Inc.) followed by 6 × 30 s sonication at 50% duty cycle and output 8 on a Branson Sonicator. Lysed sample was centrifuged at 4 °C for 30 min at 20,000 rpm in a Beckman 45Ti rotor (Beckman Coulter 339160). The supernatant was loaded onto a 30 mL hydroxyapatite (HA) column (Type II 20 μM HA; Bio-Rad 1572000) pre-equilibrated with 20 mM $KPO_4$, pH 6.8. The column-bound sample was washed for 6 column volumes (CV) in 20 mM $KPO_4$ pH 6.8, 1 M NaCl, and 5 mM β-me, then eluted with 20 mM $KPO_4$ pH 6.8, 3.5 M NaCl, and 5 mM β-me. The eluate was dialyzed into S-Column Buffer A (10 mM Tris-HCl pH 7.4, 0.75 M NaCl, 10 mM β-me, and 0.5 mM EDTA). The dialyzed protein was loaded onto a 1 mL HiTrap SP HP cation exchange column (Cytiva 17115101), washed with 20 CV S-Column Buffer A, followed by a 10 CV wash with 37% S-Column Buffer B (10 mM Tris pH 7.4, 0.5 mM EDTA, 10 mM β-me, 2 M NaCl), and eluted over a linear gradient from 37 to 100% S-Column Buffer B. Fractions containing tetramer were pooled, concentrated with an Amicon Ultra Centrifugal Filter (10 kDa MWCO; Millipore Sigma UFC801008), aliquoted, flash-frozen in liquid nitrogen, and stored at −80 °C.

## Chromatin bead reconstitution

Biotinylated 18 × 601 DNA array was purified as described in Guse et al, 2012. Briefly, puC18 with 18 repeats of the "601" nucleosome positioning sequence (Lowary and Widom, 1998) was purified via the Gigaprep kit (Qiagen 12191) from STBL2 bacteria grown in Luria Broth (LB) media. The 18 × 601 plasmid was restriction digested with EcoRI, XbaI, DraI, and HaeII overnight, and purified by polyethylene glycol (PEG) precipitation, using incremental increases of 0.5% PEG (from 4.5 to 10%) with 10 min centrifugation at $5000 \times g$ between each

increase. Precipitated and digested $18 \times 601$ DNA was dialyzed overnight into TE buffer (10 mM Tris, pH 8, and 0.5 mM EDTA) and concentrated by ethanol precipitation. The $18 \times 601$ DNA was biotinylated by filling EcoRI overhangs with α-thio-dCTP, α-thio-dGTP, α-thio-dTTP (ChemCyte CC-3002-1, CC-3003-1, CC-3004-1), and biotin-14-dATP (Invitrogen 19524016) using Klenow fragment 3'–5' exo- (New England BioLabs M0212L). Biotinylation of the $18 \times 601$ array was verified by combining 500 ng biotinylated $18 \times 601$ with 2.5 mM NaCl and 1 µg Streptavidin-Alexa 647 (Invitrogen S32357), and electrophoresing the sample on a 0.7% agarose gel to assay migration.

In total, $18 \times 601$ reconstituted chromatin was prepared via salt dialysis of $18 \times 601$ biotinylated array with purified, recombinant nucleosomes as described in Guse et al, 2012. DNA, H2A/H2B dimer, and either H3/H4 or xlCA/H4 tetramer were combined in high-salt buffer (2 M NaCl, 10 mM Tris pH 7.5, 0.25 mM EDTA) and dialyzed over 67 h into low-salt buffer (2.5 mM NaCl, 10 mM Tris pH 7.5, 0.25 mM EDTA). H3/H4 or xlCA/H4 tetramers were added at a $1.2 \times 601$ and $1.8 \times 601$ positioning sequence ratio (respectively), and H2A/H2B dimers were added at a $2.2 \times 601$ positioning sequence ratio. Reconstituted chromatin was verified by AvaI digestion of the $18 \times 601$ chromatin array overnight, followed by SyBr Gold (Life Technologies S11494) staining of a 5% acrylamide native PAGE gel.

Biotinylated $18 \times 601$ chromatin array was bound to streptavidin-coated M-280 Dynabeads (Invitrogen 11205D) washed in bead buffer (50 mM Tris pH 7.4, 75 mM NaCl, 0.25 mM EDTA, 0.05% Triton X-100, 2.5% polyvinyl alcohol). Chromatin was bound at 2.6 fmol array per microgram of bead for 60 min at room temperature with light shaking. Excess chromatin was removed by multiple washes in bead buffer, and chromatin arrays were incubated in rabbit reticulocyte lysate with in vitro translated protein (see below).

### In vitro translation and chromatin binding assays

In vitro translated protein was produced in rabbit reticulocyte lysate using the Promega SP6 TNT Quick Coupled Transcription/Translation System (Promega L2080).

To assay binding of in vitro transcribed/translated protein to chromatin arrays, rabbit reticulocyte lysate containing the protein of interest was diluted with 5× CSF-XBT (50mM K-KEPES pH 7.7, 500 mM KCl, 250 mM sucrose, 10 mM MgCl₂, 0.5 mM CaCl₂, 25 mM EGTA, 0.25% Triton X-100) to 1× CSF-XBT. 12.5 µL of diluted protein-containing lysate was added to 1.5 µL chromatin-coated beads and incubated at 21 °C for 1 h. Beads were magnetically isolated and washed 3x with 1× CSF-XBT. Beads were boiled for 5 min at 95 °C in 10 µL protein sample buffer (0.05% bromophenol blue, 10% SDS, 50% glycerol, 200 mM Tris pH 6.8, 40 mM EDTA, 2.86 M β-me) and the sample was loaded onto a 20% SDS-PAGE gel. Immunoblotting was performed as outlined below.

### X. laevis egg extracts

CytoStatic Factor (CSF)-arrested (metaphase) egg extract was produced as described (Desai et al, 1999; Guse et al, 2012). Following ovulation of the Xenopus laevis frogs, eggs were washed 3–5× with MMR buffer (6 mM Na-HEPES pH 7.8, 100 mM NaCl,

2 mM KCl, 2 mM CaCl₂, 1 mM MgCl₂, 0.1 mM EDTA) and dejellied in MMR + 2% (w/v) L-cysteine for 5 min. Dejellied eggs were washed 3–5× in 1× CSF-XB buffer (10 mM K-HEPES pH 7.7, 100 mM KCl, 50 mM sucrose, 2 mM MgCl₂, 0.1 mM CaCl₂, 5mM K-EGTA), then washed into 1× CSF-XB buffer + 10 µg/mL LPC (Leupeptin/Pepstatin A/Chemostatin). Eggs were packed in a $13 \times 51$ mm polyallomer tube (Beckman Coulter, 326819) by centrifugation in an International Clinical Centrifuge (model CL no. 740250; rotor model 221) at ~1250 rpm for 30 s, then ~1900 rpm for 15 s at room temperature. Excess buffer was removed, and packed eggs were crushed by centrifugation in a SW55Ti rotor (Beckman Coulter 342196) for 15 min at 10,000 rpm (16 °C). The soluble cytoplasmic fraction was removed by side-puncture of the tube with a 16 G, 1.5in needle and supplemented with 10 µg/ml LPC, 10 µg/ml cytochalasin D, 50 mM sucrose, and energy mix (7.5 mM creatine phosphate, 1 mM ATP, and 1 mM MgCl₂).

### Immunodepletions

Immunodepletion of endogenous M18BP1 and CENP-C from Xenopus laevis egg extract was performed as described (Moree et al, 2011). Protein A Dynabeads (Invitrogen, 10001D) were washed in 3x CSF-XBT (10 mM K-HEPES pH 7.7, 100 mM KCl, 50 mM sucrose, 2 mM MgCl₂, 0.1 mM CaCl₂, 5 mM K-EGTA, 0.05% Triton X-100) and coupled to the antibody for 60 min at 4 °C on a rotator. To immunodeplete CENP-C from 100 µL of egg extract, 5 µg of anti-xlCENP-C antibody (rabbit, raised and purified against xlCENP-C[207-296] (Milks et al, 2009)) was bound to 33 µL of Protein A Dynabeads. To immunodeplete M18BP1 from 100 µL of egg extract, 5 µg of anti-xlM18BP1 (rabbit, raised against GST-xlM18BP1-L[161-415] and purified against xlM18BP1.S[161-375] (Moree et al, 2011)) was bound to 33 µL of Protein A Dynabeads. For the mock depletion of 100 µL of egg extract, 5 µg of whole rabbit IgG (Jackson ImmunoResearch Laboratories, Inc. 011-000-003) was bound to 33 µL of Protein A Dynabeads. After coupling to the antibody, beads were washed 3x in CSF-XBT and resuspended in egg extract. The extract was depleted for 1 h at 4 °C on a rotator, and the beads were removed by 3 × 5 min magnetic pull-downs. Immunodepletion of the egg extract was verified by western blotting.

### CENP-A assembly assays

CENP-A assembly on sperm chromatin was performed as described (Moree et al, 2011; Westhorpe et al, 2015). V5-CENP-A mRNA and HJURP mRNA for use in egg extracts were produced using the Invitrogen mMessage mMachine SP6 Transcription kit (Fisher Scientific, AM1340) following the manufacturer's instructions, except 3 µg of pCS2+ plasmid was linearized via NotI digestion for the transcription reaction. Following transcription, mRNA was purified using RNeasy mini columns (Qiagen 74104) according to manufacturer instructions.

CENP-A assembly on sperm chromatin was assayed by setting up 30 µL assembly reactions containing 25 ng/µL V5-CENP-A mRNA, 40 ng/µL HJURP mRNA, 3 µL FLAG-M18BP1-L IVT protein, and 25 µL of M18BP1 and/or CENP-C immunodepleted, CSF-arrested egg extract. CENP-N localization on sperm chromatin was assayed by setting up 30 µL assembly reactions containing 28 µL of CENP-C immunodepleted, CSF-arrested egg extract. RNA and/or IVT-supplemented or unsupplemented egg extract was incubated at

16–18 °C for 30 min (flicking tubes every 15 min) to allow RNA translation to occur. The translation reaction was terminated by the addition of 0.1 mg/ml cycloheximide, and reactions were released into interphase by the addition of 750 μM CaCl₂. Egg extract reactions were supplemented with 3000 demembranated sperm chromatin per μL. Reactions were incubated at 16–18 °C for 75 min, then diluted into 1 mL dilution buffer (BRB-80 [80 mM K-PIPES pH 6.8, 1 mM MgCl₂, 1 mM EGTA], 150 mM KCl, 0.5% Triton X-100, 30% glycerol) and incubated on ice for 5 min. To each reaction, 1 mL of fixation buffer (dilution buffer + 4% formaldehyde) was added, and reactions were layered onto a cushion of BRB-80 + 40% glycerol in 15 mL glass centrifuge tubes (Kimble 45500-15). Sperm nuclei were spun down onto acid-washed, poly-L-lysine-coated coverslips via centrifugation at 3500 rpm for 20 min and then further processed for immunofluorescence.

## Immunofluorescence preparation

Coverslips were blocked in Antibody Dilution Buffer (AbDil) (20 mM Tris-HCl, pH 7.4, 150 mM NaCl with 0.1% Triton X-100, and 2% bovine serum albumin) for 30 min. Coverslips were exposed to primary antibody diluted in AbDil for 30 min (1 μg/mL rabbit anti-xlCENP-A (raised and purified against xlCENP-A$^{1-50}$ (Milks et al, 2009)), 2 μg/mL rabbit Alexa-594 anti-xlCENP-A (raised and purified against xlCENP-A$^{1-50}$ and directly labeled with Alexa Fluor 594, invitrogen A10239)), 5 μg/mL mouse anti-FLAG (Millipore Sigma F1804), 1.5 μg/mL rabbit anti-xlCENP-N (raised and purified against xlCENP-N peptide acetyl-CPHKARNSFKI-TEKR-amide, (Moree et al, 2011)), and/or 2 μg/mL mouse anti-V5 (ThermoFisher R960-25)). Coverslips were washed 3× in AbDil, then exposed to secondary antibody diluted in AbDil for 30 min (1.5 μg/mL Alexa 647 goat anti-mouse IgG (Invitrogen A21236), 1.5 μg/mL Alexa 568 goat anti-rabbit IgG (Invitrogen A11011), or 1.5 μg/mL Alexa 488 goat anti-rabbit IgG (Invitrogen A11034)). Coverslips exposed to both rabbit Alexa-594 anti-xlCENP-A and rabbit anti-xlCENP-N antibodies were first exposed to rabbit anti-xlCENP-N, then Alexa 488 goat anti-rabbit IgG, then blocked with 200 μg/mL whole rabbit IgG (Jackson ImmunoResearch Laboratories, Inc. 011-000-003), and finally exposed to rabbit Alexa-594 anti-xlCENP-A. Coverslips were washed 3× in AbDil, then stained for 5 min with 10 μg/mL Hoechst in AbDil (ThermoFisher H3570) for DNA, and washed 2× in AbDil and 2× in 1× Phosphate-Buffered Saline (PBS). Coverslips were mounted with mounting media (90% glycerol, 10 mM Tris-HCl, pH 8.8, 0.5% p-phenylene-diamine) and sealed with clear nail polish and stored at −20 °C.

## Image acquisition and processing

Imaging was performed on a Nikon Eclipse Ti2 inverted microscope (Nikon TI2-LA-FL-2) with a CrestOptic X-Light V3 Confocal Unit (Nikon) and a Celesta Light Engine (Lumencor), controlled via NIS-Elements 6.10.01 software (Nikon). Images of sperm nuclei were acquired with a Nikon Plan Apo 60×A/1.40 oil immersion lens. Images were acquired using an Electron Multiplying Charge-Coupled Device (EMCCD) camera (iXon Life 897) and digitized to 16 bits. Z sections were taken at 0.2 μm intervals across a span of 8 μm. Displayed images of sperm nuclei are maximum intensity projections of z-stacks with brightness adjusted uniformly within a figure to facilitate viewing.

## Immunoblotting

Samples were resolved by SDS-PAGE and transferred onto a nitrocellulose membrane (Amersham GE10-6000-00). All samples were transferred in CAPS transfer buffer (10 mM 3-(cyclohexyla-mino)-1-propanesulfonic acid, pH 11.3; 0.1% SDS; 20% methanol). Samples containing 0.1 μL IVT protein and 1uL Xenopus egg extract were loaded per lane.

Membranes were blocked in 4% dry milk in TBSTx (20 mM Tris, 150 mM NaCl, 0.1% Tween-20) and probed with primary antibody. Primary antibodies used were 2 μg/mL rabbit anti-xlM18BP1 (raised against GST-xlM18BP1-L$^{161-415}$ and purified against xlM18BP1-S$^{161-375}$ (Moree et al, 2011), 1.5 μg/mL rabbit anti-xlCENP-C (raised and purified against xCENP-C$^{207-296}$ (Milks et al, 2009), 1.5 μg/mL mouse anti-DM1α (tubulin; Sigma T6199), and 1 μg/mL mouse anti-myc clone 4A6 (Sigma 05-724MG). Detection was performed using horseradish peroxidase-conjugated goat-anti-mouse or goat-anti-rabbit secondary antibodies (Bio-Rad 1706516, 1706515) followed by chemiluminescence (Pierce ECL Western Blotting Substrate, Thermofisher 32106). Chemilumines-cent membranes were exposed to X-ray film for 30–300 s and developed with a Konica SRX-101A film processor.

## Mass spectrometry

To assay phosphorylation of MBP-M18BP1-L$^{747-944}$ and MBP-M18BP1-S$^{750-917}$, purified protein was added to 200 μL interphase or metaphase egg extract at 500 nM and incubated for 1 h at 20 °C. Reactions were diluted twofold in CSF-XBT, and MBP-tagged protein was pulled out by incubation for 1 h at 4 °C using 10 μg rabbit anti-MBP antibody (generated by Cocalico Biologicals and purified from rabbit serum) coupled to 40 μL Protein A Dynabeads (Invitrogen 10002D). After extensive washing in CSF-XBT, beads were incubated in sample buffer (50 mM Tris, pH 6.8, 15 mM EDTA, 1 M β-mercaptoethanol, 3.3% SDS, 10% glycerol, and 1 mg/mL Bromophenol blue) and separated via a 10% SDS-PAGE gel, stained with GelCode Blue Stain Reagent (ThermoFisher Scientific 24590). The protein band was excised from the gel and mass spectrometry was performed by the Taplin Mass Spectrometry facility, as previously described (French et al, 2017), however chymotrypsin was used instead of trypsin. The abundance of each modification was determined by summing the peak max intensity of unique peptides including each modified residue and dividing by the sum of the max intensities of all peptides containing that residue, using a Matlab script (https://gitlab.com/straightlab/mass_spec).

## Protein alignment

Protein sequences homologous to X. laevis M18BP1-L$^{758-779}$ were identified using NCBI PSI-BLAST. Multiple sequence alignment was performed in SnapGene using the MAFFT algorithm (Katoh et al, 2002).

## Quantification and statistical analysis

Image analysis of Xenopus laevis sperm was performed using custom software as described (Moree et al, 2011) and publicly available at https://github.com/cjfuller/imageanalysistools. To iden-tify centromeres, images were normalized by median filtering the

image and dividing the original image by the filtered intensity value. The channel was thresholded with the indicated centromere marker (generally CENP-A) to generate a centromere mask, and then filtered by size to remove regions larger or smaller than centromeres (using a minimum cut-off of 5 pixels and a maximum cut-off of 25 pixels). Masks were then manually inspected to remove remaining non-centromere regions. Mean pixel intensity per centromere region was measured for channels of interest using maximum intensity projections. Values for each centromere region are plotted in each dot plot as circles for every experiment and condition and normalized to the WT, with the mean of each biological replicate shown as triangles, and the overall mean and standard error shown with lines. For sperm, at least 100 centromeres among at least 10 nuclei were counted for each condition in each replicate in each experiment. All M18BP1 localization experiments were repeated for three biological replicates ($n = 3$), all CENP-A assembly experiments were repeated for four biological replicates ($n = 4$), and the CENP-N localization experiment was repeated for three biological replicates ($n = 3$).

Western blots were quantified using ImageJ (National Institutes of Health). A rectangle of fixed size was drawn around every band (both bound protein and loading control), and the mean grey value was measured. Background was calculated by taking the mean grey value for a region of the same size below each band. Background value was subtracted from the values measured for each band, then each band was first normalized to its respective loading control, and finally normalized to an average of the values for the bands representing WT M18BP1 binding to CENP-A.

All plotting and statistics were performed using GraphPad Prism. Every relevant experiment was analyzed using a Welch's unpaired two-tailed $t$ test on the means of the $n = 3$ or $n = 4$ biological replicates.

## Data availability

All source data is available at the EMBL-EBI BioStudies site with accession number S-BIAD2416. To access source data, please use the link https://www.ebi.ac.uk/biostudies/bioimages/studies/S-BIAD2416.

The source data of this paper are collected in the following database record: biostudies:S-SCDT-10_1038-S44319-026-00714-7.

## Peer review information

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

## Acknowledgements

The authors are thankful to Pragya Sidhwani for feedback on the manuscript. We also thank the Taplin Mass Spectrometry Facility at Harvard University for their assistance. This work was supported by the grant of National Institutes of Health R01 GM074728 to Aaron Straight; Jacob Price Schwartz and Rae Brown were supported by National Institutes of Health 5 T32 GM007276; and Rae Brown was supported by a National Science Foundation Graduate Research Fellowship DGE-1656518.

## Author contributions

**Rae R Brown**: Conceptualization; Resources; Data curation; Formal analysis; Funding acquisition; Validation; Investigation; Visualization; Methodology; Writing—original draft; Writing—review and editing. **Jacob P Schwartz**: Investigation; Methodology. **Lyin Ghadri**: Investigation. **Aaron F Straight**: Conceptualization; Resources; Data curation; Formal analysis; Supervision; Funding acquisition; Validation; Visualization; Methodology; Project administration; Writing—review and editing.

Source data underlying figure panels in this paper may have individual authorship assigned. Where available, figure panel/source data authorship is listed in the following database record: biostudies:S-SCDT-10_1038-S44319-026-00714-7.

## Disclosure and competing interests statement

The authors declare no competing interests.

# Expanded View Figures

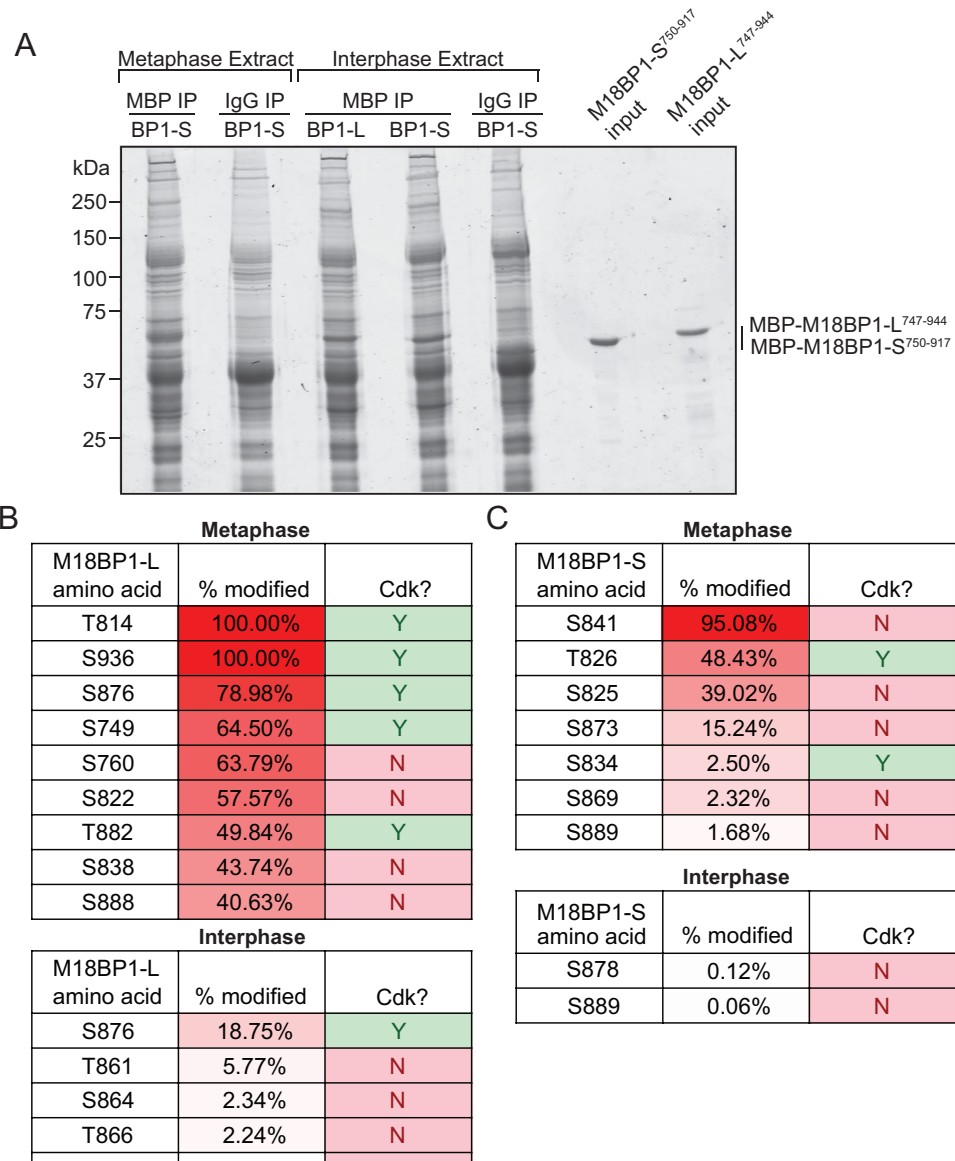

**Figure EV1. Mass spectrometry of M18BP1 from interphase or metaphase egg extract.**

(A) SDS-PAGE gel of MBP-M18BP1-S$^{750-917}$ and MBP-M18BP1-L$^{747-944}$ samples submitted for mass spectrometry. Metaphase egg extract was supplemented with *X. laevis* MBP-M18BP1-S$^{750-917}$ and interphase egg extract was supplemented with *X. laevis* MBP-M18BP1-S$^{750-917}$ and MBP-M18BP1-L$^{747-944}$, then immunoprecipitated and submitted for mass spectrometry. Mock IgG immunoprecipitation samples are shown, as well as input MBP-M18BP1-S$^{750-917}$ and MBP-M18BP1-L$^{747-944}$ protein. (B) Metaphase table adapted from French et al, 2017, displaying the nine most abundant phosphorylation events of M18BP1-L in metaphase egg extract detected with mass spectrometry. Interphase table displays data collected in this manuscript, for all phosphorylation events of M18BP1-L in interphase egg extract detected with mass spectrometry. Estimated abundance of each residue is shown, and green highlighting indicates presence of a Cdk consensus motif (S/T-P). (C) Tables displaying the phosphorylation events of M18BP1-S in metaphase or interphase egg extract detected with mass spectrometry. Estimated abundance of each residue is shown, and green highlighting indicates presence of a Cdk consensus motif (S/T-P).

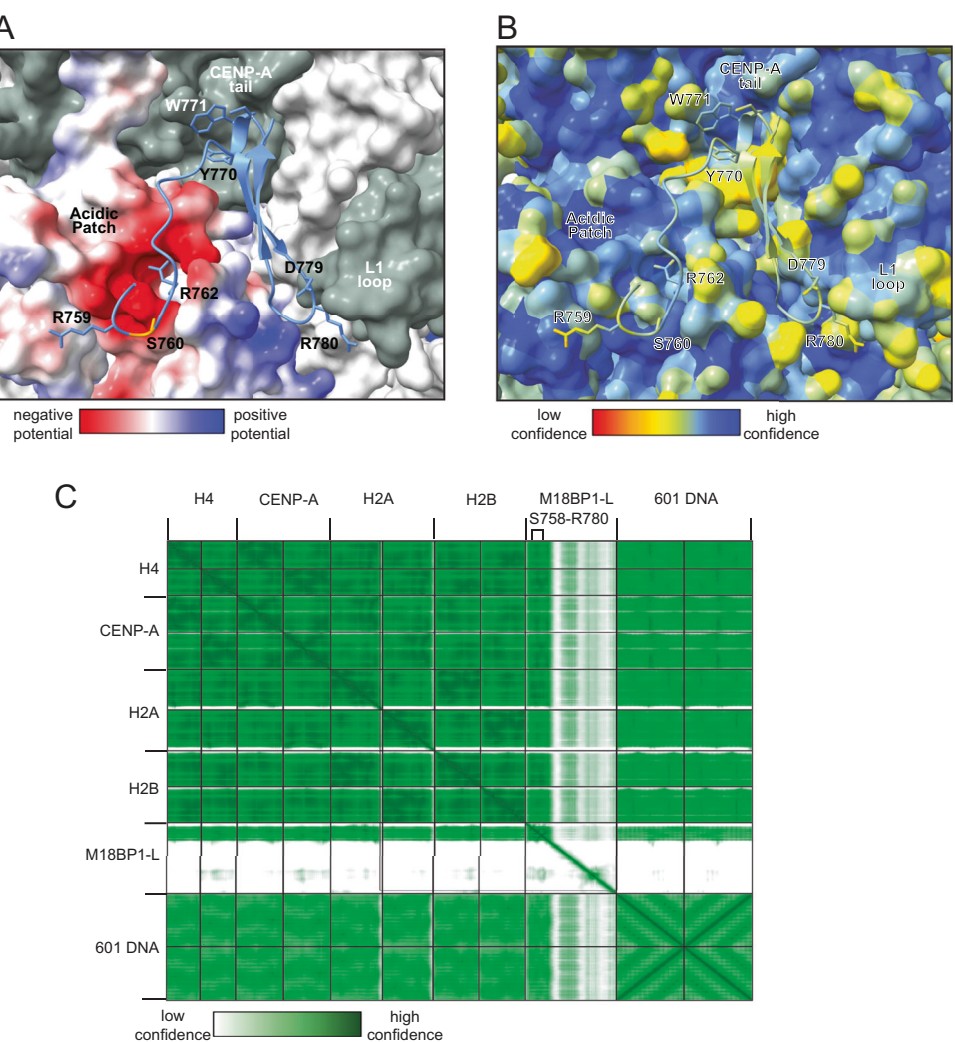

**Figure EV2. AlphaFold model of M18BP1-L bound to CENP-A.**

(A) AlphaFold structural model of the *X. laevis* CENP-A nucleosome bound to M18BP1-L[758-789]. The surface model of the C-term tail and L1 loop of the CENP-A nucleosome is shown in dark gray and the surface model of the acidic patch of H2A/H2B is colored by electrostatic potential, with red depicting negative potential and blue depicting positive potential. M18BP1-L is shown in blue, with the conserved residues shown to interact with the CENP-A nucleosome in *G. gallus* labeled (Jiang et al, 2023) and residue S760 highlighted in orange. (B) The same AlphaFold structural model as Fig. EV3A, however the model is colored according to the pLDDT local confidence value (low confidence is red, and high confidence is blue). (C) Predicted Aligned Error (PAE) plot depicting global confidence of the AlphaFold structural model depicted in Fig. 1B and Fig. EV2A,B. Location of the individual proteins and 601 DNA sequence are depicted along the top and left-hand side of the plot. The residues S758-R780 of M18BP1-L are labeled along the top of the plot.

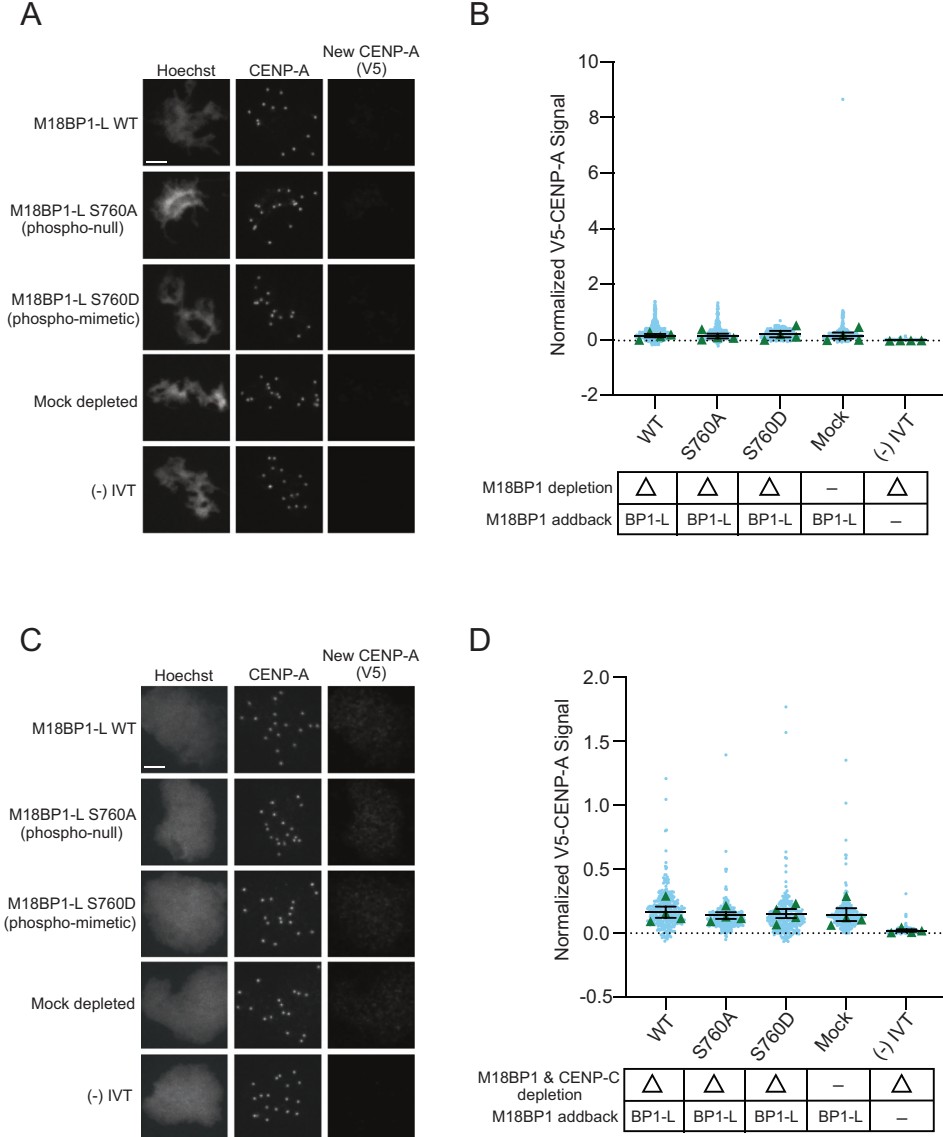

**Figure EV3. M18BP1-L phospho-mutants do not support metaphase CENP-A assembly.**

(A) Representative immunofluorescence images of new V5-CENP-A assembly in metaphase extract immunodepleted of endogenous M18BP1 then supplemented with full-length WT or mutant FLAG-M18BP1-L or a mock depletion or (−) IVT control (indicated on left). Labeling for DNA (Hoechst), total CENP-A, and new CENP-A (V5) is indicated above the image. Scale bar is 5 μm. (B) Quantification of new V5-CENP-A with controls (indicated below) in metaphase egg extract immunodepleted of endogenous M18BP1. M18BP1 depletion and addback condition is indicated in the bottom table. The signal is normalized to the WT FLAG-M18BP1-L addback condition. Error bars represent SEM of four independent replicates ($n = 4$) with green triangles displaying the mean of each replicate and blue circles representing each individual centromere. (C) Representative immunofluorescence images of new V5-CENP-A assembly in metaphase extract immunodepleted of endogenous CENP-C and M18BP1 then supplemented with full-length WT or mutant FLAG-M18BP1-L or a mock depletion or (−) IVT control (indicated on left). Labeling for DNA (Hoechst), total CENP-A, and new CENP-A (V5) is indicated above the image. Scale bar is 5 μm. (D) Quantification of new V5-CENP-A with controls (indicated below) in metaphase egg extract immunodepleted of endogenous CENP-C and M18BP1. CENP-C and M18BP1 depletion and addback condition is indicated in the bottom table. The signal is normalized to the WT FLAG-M18BP1-L addback condition. Error bars represent SEM of four independent replicates ($n = 4$) with green triangles displaying the mean of each replicate and blue circles representing each individual centromere.

