## [Peer Review File · EMBO Reports]

Phosphorylation of Xenopus M18BP1 governs centromeric localization and CENP-A nucleosome assembly

Rae Brown, Jacob Schwartz, Lyin Ghadri, and Aaron Straight

Corresponding author(s): Aaron Straight (astraight@stanford.edu)

Review Timeline:

Submission Date:	17th Jul 25
Editorial Decision:	15th Aug 25
Revision Received:	2nd Dec 25
Editorial Decision:	9th Jan 26
Revision Received:	10th Jan 26
Accepted:	29th Jan 26

Editor: Deniz Senyilmaz Tiebe / Esther Schnapp

Transaction Report:

Dear Aaron,

Thank you for transferring your manuscript to EMBO Reports, which was now seen by three referees, whose reports are copied below.

Referees express interest in the proposed regulation of M18BP1 centromeric localization and CENP-A assembly in *X. laevis*. However, they also raise some concerns that need to be addressed to consider publication here.

I find the reports informed and constructive, and believe that addressing the concerns raised will significantly strengthen the manuscript. As the reports are below, and I think all points need to be addressed, I will not detail them here. Please contact me if you have questions or comments regarding the revision for further discussion (also by video chat).

Should you be able to address all referee concerns, we would like to invite you to revise your manuscript with the understanding that the referee concerns (as in their reports) must be fully addressed and their suggestions taken on board. Please address all referee concerns in a complete point-by-point response. Acceptance of the manuscript will depend on a positive outcome of a second round of review. It is EMBO reports policy to allow a single round of major experimental revision only and acceptance or rejection of the manuscript will therefore depend on the completeness of your responses included in the next, final version of the manuscript.

We realize that it is difficult to revise to a specific deadline. In the interest of protecting the conceptual advance provided by the work, we recommend a revision within 3 months. Please discuss the revision progress ahead of this time with me if you require more time to complete the revisions, or if you have questions or comments regarding the revision (also by video chat).

1. A data availability section providing access to data deposited in public databases is missing (where applicable).
2. Your manuscript contains statistics and error bars based on $n=2$. Please use scatter plots in these cases.

You can submit the revision either as a Scientific Report or as a Research Article. For Scientific Reports, the revised manuscript can contain up to 5 main figures and 5 Expanded View figures, and it should not exceed 27000 characters. If the revision leads to a manuscript with more than 5 main figures it will be published as a Research Article. In this case the Results and Discussion section should be separate. If a Scientific Report is submitted, these sections have to be combined. This will help to shorten the manuscript text by eliminating some redundancy that is inevitable when discussing the same experiments twice. In either case, all materials and methods should be included in the main manuscript file.

4) a .docx formatted letter INCLUDING the reviewers' reports and your detailed point-by-point responses to their comments. As part of the EMBO publication's Transparent Editorial Process, EMBO reports publishes online a Review Process File (RPF) to accompany accepted manuscripts. This File will be published in conjunction with your paper and will include the referee reports,

your point-by-point response and all pertinent correspondence relating to the manuscript.

<https://www.embopress.org/page/journal/14693178/authorguide#transparentprocess>

5) a complete author checklist, which you can download from our author guidelines

<https://www.embopress.org/page/journal/14693178/authorguide>. Please insert information in the checklist that is also reflected in the manuscript. The completed author checklist will also be part of the RPF.

6) Please note that all corresponding authors are required to supply an ORCID ID for their name upon submission of a revised manuscript (<<https://orcid.org/>>). Please find instructions on how to link your ORCID ID to your account in our manuscript tracking system in our Author guidelines

<<https://www.embopress.org/page/journal/14693178/authorguide#authorshipguidelines>>

7) Before submitting your revision, primary datasets produced in this study need to be deposited in an appropriate public database (see <https://www.embopress.org/page/journal/14693178/authorguide#datadeposition>). Please remember to provide a reviewer password if the datasets are not yet public. The accession numbers and database should be listed in a formal "Data Availability" section placed after Materials & Method (see also

<https://www.embopress.org/page/journal/14693178/authorguide#datadeposition>). Please note that the Data Availability Section is restricted to new primary data that are part of this study. * Note - All links should resolve to a page where the data can be accessed. *

Additional information on source data and instruction on how to label the files are available:

<https://www.embopress.org/page/journal/14693178/authorguide#sourcedata>

9) Our journal encourages inclusion of *data citations in the reference list* to directly cite datasets that were re-used and obtained from public databases. Data citations in the article text are distinct from normal bibliographical citations and should directly link to the database records from which the data can be accessed. In the main text, data citations are formatted as follows: "Data ref: Smith et al, 2001" or "Data ref: NCBI Sequence Read Archive PRJNA342805, 2017". In the Reference list, data citations must be labeled with "[DATASET]". A data reference must provide the database name, accession number/identifiers and a resolvable link to the landing page from which the data can be accessed at the end of the reference. Further instructions are available at <http://www.embopress.org/page/journal/14693178/authorguide#referencesformat>

10) Regarding data quantification (see Figure Legends:

<https://www.embopress.org/page/journal/14693178/authorguide#figureformat>)

12) Please also note our reference format:

13) All Materials and Methods need to be described in the main text using our 'Structured Methods' format, which is required for all research articles. According to this format, the Methods section includes a Reagents and Tools Table (listing key reagents, experimental models, software and relevant equipment and including their sources and relevant identifiers) followed by a Methods and Protocols section describing the methods using a step-by-step protocol format. The aim is to facilitate adoption of the methodologies across labs. More information on how to adhere to this format as well as a downloadable template (.docx) for the Reagents and Tools Table can be found in our author guidelines:

I look forward to seeing a revised version of your manuscript when it is ready. Please let me know if you have questions or comments regarding the revision.

Kind regards,

Deniz

Deniz Senyilmaz Tiebe, PhD
Senior Scientific Editor
EMBO Reports

Referee #1:

CENP-A nucleosomes define the location of the centromere across many organisms and are critical for accurate chromosome segregation. CENP-A is a histone H3 variant. Unlike new histone H3 nucleosome assembly, new nucleosome assembly of the centromeric histone H3 variant CENP-A is uncoupled from DNA replication and occurs in G1 in vertebrates. The site of new CENP-A nucleosome assembly in each G1 is coupled to the site of the existing centromere. The mechanism by which this occurs is not entirely clear but requires the Mis18 complex and the HJURP CENP-A specific chaperone. The process of centromere assembly been studied in several model system, which have distinct and important differences. While Mis18 complex does not directly bind to CENP-A in humans to regulate its deposition, in *Xenopus laevis*, M18BP1 binds to CENP-A in interphase but not in metaphase, thereby temporally restricting CENP-A deposition. However, the mechanism for regulation of the M18BP1-CENP-A interaction is not clear. Brown et al investigate the regulation behind the restriction of CENP-A assembly in *Xenopus laevis*. The authors propose that M18BP1-L S760 phosphorylation regulates its binding to CENP-A nucleosomes. Moreover, they repeat a previous demonstration that the CENP-C pathway and CENP-A binding pathways are separable. On the whole, the experiments support the proposed model and provide clear evidence for a role of S760 in regulation of M18BP1-L binding to CENP-A. Moreover, they define the biological importance of co-targeting through CENP-C and CENP-A binding.

1. The dynamic nature of phosphorylation at S760 is assumed in the manuscript. The original manuscript (French et al, 2017) identified phosphorylation at S760 in metaphase egg extracts. This is consistent with an inhibitory function of M18BP1-L recruitment. Is S760 phosphorylation lost in interphase? And what kinase is predicted to be responsible for phosphorylation? These issues should at least be discussed more fully in the manuscript.

2. The authors should provide additional metrics for the modeling that is presented in figure 5B. How confident are the positional assignments of the amino acids that mediate the interaction between CENP-A and M18BP1-L? What are the ipTM values for the protein complexes. How much does CENP-A binding contribute to the strength of the proposed complex.

3. Based on the predicted structure, the authors propose that CENP-N may act to shield the CENP-A binding domain from Mis18 interaction in order alter the CENP-A binding requirement in mitosis. Directly demonstrating that CENP-N alters access of M18BP1 to CENP-A during mitosis would be a good extension of the current work.

Referee #2:

This study builds on previous work from the group investigating the cell cycle-dependent dynamics and regulation of the CENP-

A assembly machinery in *Xenopus* as a model system. This is an important area of research, as it addresses how centromeric chromatin is faithfully maintained across cell divisions. The key factors involved include HJURP, the dedicated CENP-A chaperone, as well as CENP-C and the Mis18 complex, both of which have been shown to contribute to HJURP recruitment during interphase, thereby enabling CENP-A incorporation.

As described in their previous study, *Xenopus* expresses two M18BP1 isoforms, S and L. Both isoforms can bind CENP-A nucleosomes during interphase through a conserved recognition motif and are important for CENP-A loading. In metaphase, only the S isoform remains associated with centromeres, but this binding occurs via CENP-C and depends on CDK1-mediated phosphorylation of M18BP1-S. The presence of M18BP1-S in addition to modifications of HJURP inhibit HJURP localization to centromeres in metaphase, thereby inhibiting CENP-A assembly during this phase of the cell cycle.

Here the authors focus on the L isoform and investigate the role of a mitotic phosphorylation in regulating M18BP1-L binding to CENP-A nucleosomes thereby identifying a new feature that contributes to restrict CENP-A assembly to interphase.

More specifically, the authors identified a conserved phosphorylation site (S760) within the M18BP1 CENP-A recognition motif that, when phosphorylated, prevents M18BP1-L CENP-A binding *in vitro* and reduces M18BP1-L centromere localization during interphase. The effect on *de novo* CENP-A assembly was however mild but enhanced upon additional depletion of CENP-C.

Finally, the authors tested whether a S760A mutation enabled M18BP1-L localization to centromere during metaphase; however, this was not the case suggesting that additional modifications or proteins may be required.

The data are clear and well-presented and I have no technical concerns. However, I think it would help the reader if the authors provided a clearer explanation of the existing literature on the sequence differences, dynamics, regulation and role of the M18BP1-L and S isoforms, as the S isoform is only briefly mentioned in the introduction and the final results section.

In this context, I was wondering to what extent their findings explain the change in M18BP1-S targeting in metaphase by inhibiting the interaction to CENP-A? As far as the alignment is concerned the S760 is conserved in the S isoform. Is the S - isoform phosphorylated at that site? Could this modification contribute the switch in S binding to CENP-C, thereby contributing to prevent CENP-A assembly in metaphase?

On the other side, they found that the S760A mutation does not enable M18BP1-L localization to metaphase centromeres, even in the absence of CENP-C. They hypothesize that this may be due to competitive binding of other CCAN components (CENP-N) to the CENP-A nucleosome, which blocks M18BP1-L recruitment. This is an interesting hypothesis. Is it also possible that M18BP1-L is unable to bind CENP-C due to the lack of modifications that promote CENP-C interaction? This would be consistent that even in the absence of M18BP1-S, the L isoform is unable to bind centromeres in metaphase (Figure 4A), which means that L and S are not competing to bind to CENP-C. If the authors agree with this interpretation, it could be valuable to include it in the discussion.

Referee #3:

Accurate chromosome segregation relies on the assembly of the microtubule-binding kinetochore complex on the centromere, a specialised chromosomal locus epigenetically defined by the enrichment of a Histone H3 variant CENP-A containing nucleosomes. The CENP-A level at the centromere undergoes replication-induced dilution during S-phase, as, unlike for the canonical histones, CENP-A deposition is decoupled from DNA replication. To preserve centromere identity, the correct CENP-A level needs to be restored in G1 through active CENP-A deposition, mediated by the Mis18 complex in cooperation with the CENP-A specific chaperone HJURP, during each cell cycle. Hence, understanding the mechanisms contributing to the cell cycle timing of targeting the CENP-A loading machinery to the centromere and subsequent CENP-A deposition is of prime importance. Here, Rae et al., using the *Xenopus* egg extract system, define a new regulatory mechanism involving a phosphoregulatory switch mediated by M18BP1 phosphorylation, in restricting the centromere targeting of the Mis18 complex to late mitosis/early G1. This is a high-quality work with new insights. However, addressing the following points prior to publication will strengthen the conclusion and improve the overall quality of the manuscript.

Lines 101-105: "To identify residues in M18BP1-L that regulate the timing of its centromere localization, we previously mapped mitotic phosphorylation sites in its CENP-A binding domain. Of these sites, the Cdk1 consensus phosphorylations did not disrupt the binding of M18BP1-L to CENP-A nucleosomes *in vitro* (French et al. 2017). Thus, we focused our analysis on the next most highly phosphorylated residue, serine 760." Readers would benefit if a supplementary table listing the various phosphorylation sites (clearly marking - Cdk1 consensus sites and non-consensus ones) that have been identified and reported in their previous study. It is not clear if there is any speculation on the kinase responsible for the phosphorylation of S760. Also, does the M18BP1 S760 phosphorylation completely disappear during interphase?

Fig. 1. Shows that M18BP1 L1 can bind CENP-A nucleosome while S760D mutation perturbs this interaction. Moving the AlphaFold model to this figure and using the model to support the nucleosome binding data, rather than discussing it only in the discussion, might work better. Also, colouring the nucleosome based on the electrostatic surface charge potential would help readers appreciate how S760 phosphorylation might perturb the nucleosome binding.

Fig. 2B. According to the western blot shown, the S760D level in the addback experiment seems to be lower compared to either WT or S760A. It is important to address this to strengthen the conclusion that the S760D is less efficient in associating with the centromere in interphase.

Lines 173-175: "M18BP1-L does not localize to the centromere in metaphase when S760 is phosphorylated, unlike its isoform M18BP1-S (French and Straight 2019; Flores Servin, Brown, and Straight 2023). " Although the relevant papers are cited, more context and specific information here would be helpful. Does this mean M18BP1 L S760 phosphorylation has previously been characterised? Or Is this just about differential localisation of the isoforms? It is not clear from a quick look at these previous publications.

Fig. 5A. The model is too abstract. This manuscript is about the cell cycle-specific phosphorylation of M18BP1-L1 S760 as a new regulatory mechanism for the cell-cycle control of CENP-A deposition, but S760 is not mentioned in the cartoon/model. It would be more impactful if this phosphorylation along with some of the other key regulatory phosphorylations reported previously (French and Straight 2019) could be included to get a more comprehensive view of the cell-cycle control of M18BP1 centromere targeting and CENP-A deposition.

Moreover, the data presented here, while suggests that removal of M18BP1 S760 phosphorylation is required for interphase localisation (S760A associates with centromere and facilitates CENP-A deposition), does not really show that the M18BP1 S760 phosphorylation blocks its association with the centromere in metaphase (as the nonphosphorylatable (S760A) mutant also fail to associate with metaphase centromere) and as mentioned in the discussion there is likely another mechanism preventing M18BP1 to bind CENP-A nucleosome. This also needs to be conveyed in the model Fig. 5A.

Author responses are in bold green text

We appreciate the careful analysis of our manuscript by the referees. We have performed additional experimental work as well as included new text to clarify or respond to each reviewer point. The point by point response is included below.

Referee #1:

CENP-A nucleosomes define the location of the centromere across many organisms and are critical for accurate chromosome segregation. CENP-A is a histone H3 variant. Unlike new histone H3 nucleosome assembly, new nucleosome assembly of the centromeric histone H3 variant CENP-A is uncoupled from DNA replication and occurs in G1 in vertebrates. The site of new CENP-A nucleosome assembly in each G1 is coupled to the site of the existing centromere. The mechanism by which this occurs is not entirely clear but requires the Mis18 complex and the HJURP CENP-A specific chaperone. The process of centromere assembly been studied in several model system, which have distinct and important differences. While Mis18 complex does not directly bind to CENP-A in humans to regulate its deposition, in *Xenopus laevis*, M18BP1 binds to CENP-A in interphase but not in metaphase, thereby temporally restricting CENP-A deposition. However, the mechanism for regulation of the M18BP1-CENP-A interaction is not clear. Brown et al investigate the regulation behind the restriction of CENP-A assembly in *Xenopus laevis*. The authors propose that M18BP1-L S760 phosphorylation regulates its binding to CENP-A nucleosomes. Moreover, they repeat a previous demonstration that the CENP-C pathway and CENP-A binding pathways are separable. On the whole, the experiments support the proposed model and provide clear evidence for a role of S760 in regulation of M18BP1-L binding to CENP-A. Moreover, they define the biological importance of co-targeting through CENP-C and CENP-A binding.

1. The dynamic nature of phosphorylation at S760 is assumed in the manuscript. The original manuscript (French et al, 2017) identified phosphorylation at S760 in metaphase egg extracts. This is consistent with an inhibitory function of M18BP1-L recruitment. Is S760 phosphorylation lost in interphase? And what kinase is predicted to be responsible for phosphorylation? These issues should at least be discussed more fully in the manuscript.

We performed phosphoproteomics on M18BP1-L/S from interphase and metaphase egg extract using mass spectrometry, and found no evidence of M18BP1-L S760 phosphorylation in interphase. We have included the data and discussion of the results in Figure EV1 and lines 106-112.

We included a new discussion of the possible kinases that are phosphorylating M18BP1-L S760 in lines 285-292. We agree that identifying the kinase will be important for future understanding of the regulatory mechanism. However, we feel that an extensive additional study to identify the kinase is outside the scope of this manuscript.

2. The authors should provide additional metrics for the modeling that is presented in figure 5B. How confident are the positional assignments of the amino acids that mediate the interaction between CENP-A and M18BP1-L? What are the ipTM values for the protein complexes. How much does CENP-A binding contribute to the strength of the proposed complex.

We have included additional metrics for the alphafold model, including mapping the pLDDT on the model and displaying the PAE plot (Fig EV2), as well as adding the ipTM and pTM values to the figure legend of Fig 1B.

3. Based on the predicted structure, the authors propose that CENP-N may act to shield the CENP-A binding domain from Mis18 interaction in order alter the CENP-A binding requirement in mitosis. Directly demonstrating that CENP-N alters access of M18BP1 to CENP-A during mitosis would be a good extension of the current work.

We do not have an antibody that is able to immunodeplete CENP-N from egg extract, and thus we could not directly test if loss of CENP-N changes metaphase M18BP1-L localization. Instead, we depleted CENP-C in Figure 5 and then assayed endogenous CENP-N localization in egg extract. In the absence of CENP-C, CENP-N is unable to localize to centromeres in metaphase (Figure 5, text lines 208-236). Thus if CENP-N is unable to bind to centromeres in metaphase in the absence of CENP-C, it cannot be occluding M18BP1 S760A localization in the results shown in Figure 4C-D.

Referee #2:

This study builds on previous work from the group investigating the cell cycle-dependent dynamics and regulation of the CENP-A assembly machinery in *Xenopus* as a model system. This is an important area of research, as it addresses how centromeric chromatin is faithfully maintained across cell divisions. The key factors involved include HJURP, the dedicated CENP-A chaperone, as well as CENP-C and

the Mis18 complex, both of which have been shown to contribute to HJURP recruitment during interphase, thereby enabling CENP-A incorporation. As described in their previous study, *Xenopus* expresses two M18BP1 isoforms, S and L. Both isoforms can bind CENP-A nucleosomes during interphase through a conserved recognition motif and are important for CENP-A loading. In metaphase, only the S isoform remains associated with centromeres, but this binding occurs via CENP-C and depends on CDK1-mediated phosphorylation of M18BP1-S. The presence of M18BP1-S in addition to modifications of HJURP inhibit HJURP localization to centromeres in metaphase, thereby inhibiting CENP-A assembly during this phase of the cell cycle.

Here the authors focus on the L isoform and investigate the role of a mitotic phosphorylation in regulating M18BP1-L binding to CENP-A nucleosomes thereby identifying a new feature that contributes to restrict CENP-A assembly to interphase. More specifically, the authors identified a conserved phosphorylation site (S760) within the M18BP1 CENP-A recognition motif that, when phosphorylated, prevents M18BP1-L CENP-A binding in vitro and reduces M18BP1-L centromere localization during interphase. The effect on de novo CENP-A assembly was however mild but enhanced upon additional depletion of CENP-C. Finally, the authors tested whether a S760A mutation enabled M18BP1-L localization to centromere during metaphase; however, this was not the case suggesting that additional modifications or proteins may be required.

The data are clear and well-presented and I have no technical concerns. However, I think it would help the reader if the authors provided a clearer explanation of the existing literature on the sequence differences, dynamics, regulation and role of the M18BP1-L and S isoforms, as the S isoform is only briefly mentioned in the introduction and the final results section.

We added a more thorough discussion of the differences between the L and S isoforms to the Discussion section (lines 293-305). We also added a synopsis model and a model in Figure 6 that better highlights the differences between the factors in the system and their cell cycle specific functions.

In this context, I was wondering to what extent their findings explain the change in M18BP1-S targeting in metaphase by inhibiting the interaction to CENP-A? As far as the alignment is concerned the S760 is conserved in the S isoform. Is the S -isoform phosphorylated at that site? Could this modification contribute the switch in S binding to CENP-C, thereby contributing to prevent CENP-A assembly in metaphase?

We think that this is a good point and one we have considered. In our mass spectrometry data we were unable to detect the peptide containing S772 in metaphase or interphase samples so could not assess whether it is also phosphorylated.

One challenge is that M18BP1-S is difficult to reliably produce by *in vitro* transcription/translation making it difficult to assay its localization or complementation in extract.

Given these limitations we can't definitively assess whether M18BP1-S S772 is phosphorylated and whether that contributes to the switch to CENP-C binding in metaphase. Our previously published data (French and Straight 2019) establishes the localization mechanism in metaphase through CENP-C binding but given the challenges in assaying M18BP1-S we feel that determining whether S772 phosphorylation contributes to the switch to CENP-C binding in metaphase is beyond the scope of this current manuscript.

On the other side, they found that the S760A mutation does not enable M18BP1-L localization to metaphase centromeres, even in the absence of CENP-C. They hypothesize that this may be due to competitive binding of other CCAN components (CENP-N) to the CENP-A nucleosome, which blocks M18BP1-L recruitment. This is an interesting hypothesis. Is it also possible that M18BP1-L is unable to bind CENP-C due to the lack of modifications that promote CENP-C interaction? This would be consistent that even in the absence of M18BP1-S, the L isoform is unable to bind centromeres in metaphase (Figure 4A), which means that L and S are not competing to bind to CENP-C. If the authors agree with this interpretation, it could be valuable to include it in the discussion.

In previous publications (French and Straight 2019; Moree et al. 2011), we showed that only full-length M18BP1-S localizes to the metaphase centromere by binding to CENP-C while M18BP1-L cannot localize to the metaphase centromere or bind to CENP-C (Moree et al. 2011). Thus as you note M18BP1-L is unable to compete for binding to CENP-C. We agree with the interpretation you highlight and we have added further discussion of the metaphase M18BP1-S vs M18BP1-L localization differences in the discussion, lines 293-305.

Referee #3:

Accurate chromosome segregation relies on the assembly of the microtubule-binding kinetochore complex on the centromere, a specialised chromosomal locus epigenetically defined by the enrichment of a Histone H3 variant CENP-A containing nucleosomes. The CENP-A level at the centromere undergoes replication-induced

dilution during S-phase, as, unlike for the canonical histones, CENP-A deposition is decoupled from DNA replication. To preserve centromere identity, the correct CENP-A level needs to be restored in G1 through active CENP-A deposition, mediated by the Mis18 complex in cooperation with the CENP-A specific chaperone HJURP, during each cell cycle. Hence, understanding the mechanisms contributing to the cell cycle timing of targeting the CENP-A loading machinery to the centromere and subsequent CENP-A deposition is of prime importance. Here, Rae et al., using the *Xenopus* egg extract system, define a new regulatory mechanism involving a phosphoregulatory switch mediated by M18BP1 phosphorylation, in restricting the centromere targeting of the Mis18 complex to late mitosis/early G1. This is a high-quality work with new insights. However, addressing the following points prior to publication will strengthen the conclusion and improve the overall quality of the manuscript.

Lines 101-105: "To identify residues in M18BP1-L that regulate the timing of its centromere localization, we previously mapped mitotic phosphorylation sites in its CENP-A binding domain. Of these sites, the Cdk1 consensus phosphorylations did not disrupt the binding of M18BP1-L to CENP-A nucleosomes in vitro (French et al. 2017). Thus, we focused our analysis on the next most highly phosphorylated residue, serine 760." Readers would benefit if a supplementary table listing the various phosphorylation sites (clearly marking - Cdk1 consensus sites and non-consensus ones) that have been identified and reported in their previous study. It is not clear if there is any speculation on the kinase responsible for the phosphorylation of S760. Also, does the M18BP1 S760 phosphorylation completely disappear during interphase?

We could not detect any phosphorylation of S760 in interphase as stated in our response to Reviewer 1. We have included the data and discussion in Figure EV1 and lines 106-112. We also added a supplementary table outlining phosphorylation sites from both the previous manuscript and this manuscript in Fig EV1.

We further discussed the possible kinases that are phosphorylating M18BP1-L S760 in lines 285-292. Please also see our response to Reviewer 1 as the consensus site does not match Cdk1 or Plk1 sites and determining the kinase we feel would require significant additional experimental work beyond this study.

Fig. 1. Shows that M18BP1 L1 can bind CENP-A nucleosome while S760D mutation perturbs this interaction. Moving the AlphaFold model to this figure and using the

model to support the nucleosome binding data, rather than discussing it only in the discussion, might work better. Also, colouring the nucleosome based on the electrostatic surface charge potential would help readers appreciate how S760 phosphorylation might perturb the nucleosome binding.

Thank you for this suggestion. We have moved the AlphaFold model to Figure 1B and discuss it in the Results section (lines 120-123) rather than the Discussion.

We also added an electrostatic surface charge potential depiction of the AlphaFold structural model in Fig EV3. We kept the model with the original colors in Fig 1B because the electrostatic potential colors make the model more difficult to read overall. But, we agree that the surface charge does provide important context, and thus placed it in the supplement.

Fig. 2B. According to the western blot shown, the S760D level in the addback experiment seems to be lower compared to either WT or S760A. It is important to address this to strengthen the conclusion that the S760D is less efficient in associating with the centromere in interphase.

While the levels of metaphase S760D in Fig 2B do appear lower than the rest of the conditions, we have found this to be due to the variability in western blotting egg extract, and in particular M18BP1, and not due to differences in addback amount. We redid the western blot to get a better quality western blot that is more representative of the levels added to egg extract, and have replaced Fig 2B.

Lines 173-175: "M18BP1-L does not localize to the centromere in metaphase when S760 is phosphorylated, unlike its isoform M18BP1-S (French and Straight 2019; Flores Servin, Brown, and Straight 2023)." Although the relevant papers are cited, more context and specific information here would be helpful. Does this mean M18BP1 L S760 phosphorylation has previously been characterised? Or Is this just about differential localisation of the isoforms? It is not clear from a quick look at these previous publications.

We edited the referenced lines to improve the clarity of the text. We have not previously characterized the localization of S760 mutants, and we were referencing the differential localization of the isoforms. The edited text is located in lines 188-189.

Fig. 5A. The model is too abstract. This manuscript is about the cell cycle-specific phosphorylation of M18BP1-L1 S760 as a new regulatory mechanism for the cell-cycle

control of CENP-A deposition, but S760 is not mentioned in the cartoon/model. It would be more impactful if this phosphorylation along with some of the other key regulatory phosphorylations reported previously (French and Straight 2019) could be included to get a more comprehensive view of the cell-cycle control of M18BP1 centromere targeting and CENP-A deposition.

Moreover, the data presented here, while suggests that removal of M18BP1 S760 phosphorylation is required for interphase localisation (S760A associates with centromere and facilitates CENP-A deposition), does not really show that the M18BP1 S760 phosphorylation blocks its association with the centromere in metaphase (as the nonphosphorylatable (S760A) mutant also fail to associate with metaphase centromere) and as mentioned in the discussion there is likely another mechanism preventing M18BP1 to bind CENP-A nucleosome. This also needs to be conveyed in the model Fig. 5A.

We have updated the model presented in the synopsis and Fig 6 (previously Fig 5A) to be clearer, and to more thoroughly and accurately convey the information. We have omitted additional regulatory modification from previous work to avoid overcomplicating the figure with information not discussed in-depth in this manuscript.

Dear Dr. Straight,

Thank you for the submission of your revised manuscript. We have now received the enclosed reports from the referees and I am happy to say that all support its publication now. Only a few editorial requests will need to be addressed before we can proceed with the official acceptance of your manuscript:

- Please add up to 5 keywords to your ms file.
- The Data Availability Statement needs to be moved to before the Acknowledgments.
- The author credits need to be removed from the ms file. All credits need to be entered during online ms submission.
- The REFERENCE format is OK, but Bibliography should be corrected to References.
- Figure 6 has panel A labeled, but not panel B, so panel A should be removed.
- It is not clear to me whether all your source data (SD) are uploaded on BioStudies? I could not find any SD with your ms file. Can you please clarify?
- Materials and Methods should be just Methods
- The Figure legends should be placed after the References, at the very end of the ms.

I would like to suggest some minor changes to the title and abstract. Please let me know whether you agree with the following:

Phosphorylation of Xenopus M18BP1 governs centromeric localization and CENP-A nucleosome assembly

Eukaryotic chromosome segregation requires attachment of chromosomes to microtubules through the kinetochore so that chromosomes can align and move in mitosis. Kinetochores assemble on the centromere which is epigenetically defined by the histone H3 variant CENtrome Protein A (CENP-A). During DNA replication CENP-A is equally divided between replicated chromatids and new CENP-A nucleosomes are re-assembled during the subsequent G1 phase. How cells regulate the cell cycle timing of CENP-A assembly is a central question in the epigenetic maintenance of centromeres. CENP-A nucleosome assembly requires the Mis18 complex (Mis18 α , Mis18 β , and M18BP1) whose localization to centromeres occurs between metaphase and G1 [OK?]. Here, we define a new regulatory mechanism that works through phosphorylation of *Xenopus laevis* M18BP1 between metaphase and interphase. Phosphorylation disrupts binding of M18BP1 to CENP-A nucleosomes in metaphase, and when removed enables M18BP1 binding to CENP-A nucleosomes in interphase. We show that this phosphorylation-dependent mechanism regulates CENP-A nucleosome assembly. We propose that the phospho-regulated binding of M18BP1 to CENP-A nucleosomes restricts new CENP-A assembly to interphase.

EMBO press papers are accompanied online by A) a short (1-2 sentences) summary of the findings and their significance, B) 2-3 bullet points highlighting key results and C) a synopsis image that is exactly 550 pixels wide and 200-600 pixels high (the height is variable). The synopsis image you sent is OK but please send us the short summary and bullet points along with the final manuscript.

Referee #1:

The reviewers have addressed my concerns from the previous round of revision and I support publication of the manuscript.

Referee #2:

The authors have satisfactorily addressed all of my concerns and comments.

Referee #3:

The authors have adequately addressed the concerns raised by this reviewer. Hence, this reviewer is happy to support the publication.

All minor editorial requests have been addressed by the authors.

Dr. Aaron Straight
Stanford University
Department of Biochemistry
279 Campus Drive
Beckman 409
Stanford, CA 94305-5307
United States

Dear Dr. Straight,

I am very pleased to accept your manuscript for publication in the next available issue of EMBO reports. Thank you for your contribution to our journal.

You may qualify for financial assistance for your publication charges - either via a Springer Nature fully open access agreement or an EMBO initiative. Check your eligibility: <https://link.springer.com/journal/44319/how-to-publish-with-us>

Yours sincerely,

>>> Please note that it is EMBO Reports policy for the transcript of the editorial process (containing referee reports and your response letter) to be published as an online supplement to each paper. If you do NOT want this, you will need to inform the Editorial Office via email immediately. More information is available here: <https://link.springer.com/partners/embo-press/editorial-policies#Peer%20review>